# MAPPING POST-TRAINING FORGETTING IN LANGUAGE MODELS AT SCALE

**Jackson Harmon**   **Andreas Hochlehnert**   **Matthias Bethge**[*]   **Ameya Prabhu**[*]
Tübingen AI Center, University of Tübingen

`https://post-forget.github.io/`

## ABSTRACT

Scaled post-training now drives many of the largest capability gains in language models (LMs), yet its effect on pretrained knowledge remains poorly understood. Not all forgetting is equal: Forgetting one fact (e.g., a U.S. president or an API call) does not "average out" when recalling another. Hence, we propose a sample-wise paradigm to measure what is forgotten and when backward transfer occurs. Our metric counts $1 \to 0$ transitions (correct before post-training, incorrect after) to quantify forgetting and $0 \to 1$ transitions to quantify backward transfer. Traditional task averages conflate these effects and obscure large changes. For multiple-choice benchmarks, we add chance-adjusted variants that subtract the expected contribution of random guessing from pre- and post-training accuracies. We apply this framework across post-training stages, model sizes, and data scales. Our large-scale analysis across nearly 30 model pairs and 100 sub-benchmarks with up to 32,768 generated tokens per sample shows that: (1) Domain-continual pretraining induces moderate forgetting with low-to-moderate backward transfer; (2) RL/SFT post-training applied to base models and instruction tuning yields moderate-to-large backward transfer on math and logic with overall low-to-moderate forgetting; (3) Applying RL/SFT to instruction-tuned models is sensitive on data scale: at small scales, both forgetting and backward transfer are small; at larger scales, effects are mixed and warrant further study with better controls; (4) Model merging does not reliably mitigate forgetting. Overall, our framework offers a practical yardstick for mapping how post-training alters pretrained knowledge at scale – enabling progress towards generally capable AI systems.

## 1 INTRODUCTION

Scaling post-training has become the dominant driver of capability gains in modern language models (LMs) (Jaech et al., 2024). Practitioners now iterate through multi-step post-training pipelines often at data scales that rival early pretraining (Tie et al., 2025). The implicit bet is that each step in the pipeline accumulates new capabilities, with dramatic improvements in areas like coding, math, tool use and safety, without sacrificing broad world knowledge. In contrast, it is considered common knowledge in continual learning that sequential training leads to catastrophic forgetting (see Table 1). We test this assumption: as we scale post-training, do we erode the very breadth of world knowledge that pretraining painstakingly compresses into the weights? If the implicit assumption does not hold, we risk trading generalist competence for narrow specialization, undermining progress toward generally capable models.

Measuring forgetting in modern post-training pipelines is tricky. Classical evaluations compare aggregate test accuracy before and after training, implicitly treating a benchmark as a single task with fungible i.i.d. samples (e.g., classifying images of cats). Pretrained knowledge violates this assumption. Knowing one U.S. president does not compensate for forgetting another; recalling a NumPy broadcasting rule does not offset losing knowledge of a specific cloud-API syntax. In short, knowledge samples are not fungible: each carries unique value for quantifying pretraining knowledge. Aggregation can hide substantial losses. Hence, we measure forgetting and backward transfer in a sample-wise manner, rather than at the task level as proposed by Lopez-Paz & Ranzato (2017).

---

[*]Equal Supervision

Specifically, we define *forgetting* as items that are answered correctly before a post-training stage but incorrectly afterward (the $1 \rightarrow 0$ transitions), and *backward transfer* as items that are answered incorrectly before but correctly after post-training (the $0 \rightarrow 1$ transitions). A further complication is that most knowledge-intensive LLM evaluation benchmarks are multiple-choice. Random guessing inflates accuracy and can create illusory transitions: an apparent "$1 \rightarrow 0$" may simply be a lucky guess that later becomes an incorrect answer, even when the underlying knowledge did not change; likewise for $0 \rightarrow 1$ transitions. When the answer is only among few options (e.g., 4), performance by random guessing can account for a substantial share of observed transitions, distorting both level and trend estimates of forgetting. Thus a principled metric should (i) resolve outcomes at the *item* level and (ii) explicitly correct for chance.

To account for these considerations, we introduce chance-adjusted metrics for forgetting ($F_{true}$) and backward transfer ($BT_{true}$), which correct for transitions expected under random choice. They do not need logits or repeated sampling, requiring only the number of answer choices, the sample-level accuracies, and the marginal accuracy before and after training, making them practical at scale. Intuitively, chance-adjusted forgetting asks: among items the model genuinely knew before, what fraction became wrong beyond chance? Conversely, chance-adjusted backward transfer asks: among items the model genuinely did not correctly solve, what fraction became correct beyond chance?

Our primary contribution is a large-scale study measuring forgetting caused by post-training across post-training pipelines. By evaluating the models on the same set of samples before and after each stage, we obtain a map of what was retained, what was forgotten, and where losses concentrate. We seek to answer three questions: (i) Where in the pipeline is forgetting most pronounced (e.g., instruction tuning vs. reasoning-focused training)?, (ii) What kinds of pretraining knowledge are most affected (culture vs. logic)?, and (iii) How much knowledge is forgotten or re-elicited? We have the following key findings:

> **Key Findings**
>
> - **Domain-Continual Pretraining** induces low to moderate forgetting across most categories; backward transfer is limited. Forgetting effects marginally decrease with increasing model scale.
>
> - **Instruction-Tuning and SFT/RL from base models** yield low to moderate forgetting, with spikes in the Culture and Knowledge categories, but moderate to high (for SFT/RL from Base) backward-transfer gains in the Math and Logic categories across model families; forgetting and backward transfer decrease as parameters increase. Reasoning training yields similar forgetting and larger backward transfer than instruction tuning.
>
> - **SFT/RL Reasoning Post-Training from instruct models** have data-scale dependent behaviour: For the low-data regime, it yields low forgetting and backward transfer. For the high-data regime, no dominant factor robustly explains the forgetting and backward transfer dynamics.
>
> - **Model Merging** does not reliably mitigate forgetting across post-training pipelines (yet).

Table 1: **Catastrophic forgetting literature across LLM post-training stages.** Continual learning literature indicates extensive forgetting across the post-training pipeline. However, we find far less forgetting when testing widely used post-training pipelines, indicating an important gap existing between continual learning setups and how people post-train language models.

| Stage | Name | Level | Summary |
|---|---|---|---|
| CPT (§3.1) | Investigating Continual Pretraining in LLMs: Insights and Implications (Yıldız et al., 2024) | Med | Most models show continual improvement; only Llama-2 models degrade. |
| | Examining Forgetting in Continual Pretraining of Aligned LLMs (Li & Lee, 2024a) | High | Continual pretraining degrades capabilities, alignment and alters output behavior. |

*(Continued on next page)*

*(Continued from previous page)*

| Stage | Name | Level | Summary |
|---|---|---|---|
| SFT/DPO (§3.2) | Mitigating Forgetting in LLM Supervised Fine-Tuning and Preference Learning (Fernando et al., 2024) | Low | Combining SFT and DPO sequentially leads to forgetting and a poor balance between goals ($\sim 2\%$ on MMLU). |
| SFT (§3.3) | Refine Large Language Model Fine-tuning via Instruction Vector (Jiang et al., 2024) | High | Fine-tuning on TRACE shows declines primarily from lost instruction-following ability. |
| | An Empirical Study of Catastrophic Forgetting in LLMs During Continual Fine-tuning (Luo et al., 2025) | High | Forgetting of domain knowledge, reasoning intensifies as model scale increases ($\sim 10\%$ MMLU drop). |
| | Catastrophic Forgetting in LLMs: A Comparative Analysis Across Language Tasks (Haque, 2025) | High | Severity varies by architecture and pretraining quality; some models degrade sharply while others barely change. |
| | Mitigating Catastrophic Forgetting in LLMs with Self-Synthesized Rehearsal (Huang et al., 2024) | High | Sequential fine-tuning causes major forgetting; synthetic rehearsal mitigates it. |
| RL (§3.2) | Mitigating the Alignment Tax of RLHF (Lin et al., 2024) | Med | RLHF induces forgetting ("alignment tax"); model averaging reduces it. |
| SFT/RL (§3.2) | Understanding Catastrophic Forgetting in LLMs via Implicit Inference (Kotha et al., 2024) | High | Fine-tuning skews the model's implicit task inference rather than erasing capabilities. |
| | Temporal Sampling for Forgotten Reasoning in LLMs (Li et al., 2025) | High | Fine-tuned LLMs often forget solutions they previously generated ("temporal forgetting") across sizes and methods (SFT, GRPO). |

## 2 Measuring Samplewise Forgetting and Backward Transfer

To formalize these metrics, first consider an evaluation set of $N$ multiple-choice questions with $k$ options. For each sample $i$, let $a_i^{\text{pre}}, a_i^{\text{post}} \in \{0, 1\}$ indicate correctness before and after post-training. As illustrated in Fig. 1, each sample falls into one of four quadrants based on the effect of training on a new task:

(i) Retention preserves knowledge ($1 \rightarrow 1$),
(ii) Backward Transfer improves performance ($0 \rightarrow 1$),
(iii) Forgetting reduces performance ($1 \rightarrow 0$), and
(iv) non-acquisition has no effect ($0 \rightarrow 0$).

We define sample-wise *forgetting* and *backward transfer* as the proportions of $1 \rightarrow 0$ and $0 \rightarrow 1$ flips, respectively:

$$\text{F} = \frac{1}{N} \sum_{i=1}^{N} \mathbf{1}\{a_i^{\text{pre}} = 1 \land a_i^{\text{post}} = 0\}$$

$$\text{BT} = \frac{1}{N} \sum_{i=1}^{N} \mathbf{1}\{a_i^{\text{pre}} = 0 \land a_i^{\text{post}} = 1\}$$

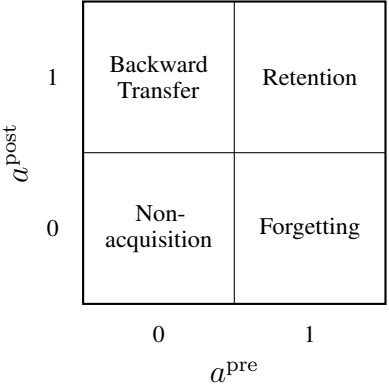

Figure 1: Each sample is assigned to one of four quadrants by correctness before and after.

However, these intuitive metrics confound genuine knowledge change with label flips caused by guessing, especially when $k$ is small. For example, two independent random binary classifiers ($k=2$) yield F $= 0.25$ because $0.5 \times 0.5 = 0.25$.

**A chance baseline for flips.** To account for guessing, we assume a simple response model: on each item the model either *knows* the answer or *guesses* uniformly among the $k$ choices. Let $\bar{a}$ be mean accuracy on a set. Then $\bar{a} = \bar{a}_{\text{true}} + x$, where $x$ is the fraction correct by chance. Since an incorrect guess occurs with probability $(k-1)/k$,

$$\frac{1 - \bar{a}}{x + (1 - \bar{a})} = \frac{k - 1}{k} \quad \implies \quad x = \frac{1 - \bar{a}}{k - 1}.$$

Figure 2: Accuracy $\bar{a}$ decomposes into true knowledge $\bar{a}_{\text{true}}$ and lucky guesses $x$.

A $1 \to 0$ flip due purely to chance requires (i) a pre-training correct guess and (ii) a post-training error (converse for backward transfer). Assuming independence between pre- and post-training guessing events,

$$\text{F}_{\text{chance}} = \underbrace{\frac{1 - \bar{a}^{\text{pre}}}{k - 1}}_{\text{correct by chance (pre)}} \cdot \underbrace{(1 - \bar{a}^{\text{post}})}_{\text{incorrect (post)}}, \qquad \text{BT}_{\text{chance}} = \underbrace{(1 - \bar{a}^{\text{pre}})}_{\text{incorrect (pre)}} \cdot \underbrace{\frac{1 - \bar{a}^{\text{post}}}{k - 1}}_{\text{correct by chance (post)}}.$$

These metrics depend only on aggregate accuracies and $k$; they require neither logits nor heavy computation.

**Chance-adjusted forgetting and backward transfer.** From these estimates we can isolate knowledge change beyond chance by subtracting the baselines from the respective forgetting/backward-transfer metrics and clipping at zero:

$$\text{F}_{\text{true}} = \max(\text{F} - \text{F}_{\text{chance}}, 0), \qquad \text{BT}_{\text{true}} = \max(\text{BT} - \text{BT}_{\text{chance}}, 0).$$

For example, if accuracy drops from 80% to 70% on a 4-option MCQ test, raw forgetting is 10%, but chance-adjusted forgetting is only about 6% – showing how the correction removes the effect of lucky guesses. Clipping ensures the metric remains valid even if models perform below chance. In practice, for an accurate measure of forgetting, this metric's mean and variance statistics should be computed over multiple seeds, as described in Section B.1.

**Ceilings: how much could a model forget or improve?** Observed forgetting can be small simply because little was truly correct to begin with. The *maximum possible* forgetting equals the fraction truly correct before post-training, which we adjust for guessing and clip at 0:

$$\text{F}_{\max} = \bar{a}^{\text{pre}}_{\text{true}} = \max(\bar{a}^{\text{pre}} - x^{\text{pre}}, 0) = \max\left(\frac{k\,\bar{a}^{\text{pre}} - 1}{k - 1}, 0\right).$$

Similarly, the *maximum possible* backward transfer equals the fraction truly correct after post-training:

$$\text{BT}_{\max} = \bar{a}^{\text{post}}_{\text{true}} = \max(\bar{a}^{\text{post}} - x^{\text{post}}, 0) = \max\left(\frac{k\,\bar{a}^{\text{post}} - 1}{k - 1}, 0\right).$$

where

$$x^{\text{pre}} = \frac{1 - \bar{a}_{\text{pre}}}{k - 1}, \qquad x^{\text{post}} = \frac{1 - \bar{a}_{\text{post}}}{k - 1}$$

By construction $\text{F}_{\text{true}} \leq \text{F}_{\max}$ and $\text{BT}_{\text{true}} \leq \text{BT}_{\max}$. Reporting the adjusted metrics alongside these ceilings separates true knowledge loss/acquisition from chance and contextualizes headroom for degradation or improvement.

**Assumptions and scope.** The correction uses two assumptions: (i) when the model does not know an answer, it guesses uniformly at random; and (ii) pre- and post-training guessing events are independent. These assumptions allow dataset-level adjustments from pre- and post-training accuracies alone. Note that $\text{F}_{\text{true}}$ could quantify failure to elicit previously accessible knowledge and need not imply that the model has lost/unlearned the underlying information. Likewise, changes in $\text{BT}_{\text{true}}$ often reflect improved elicitation rather than newly acquired knowledge.

# 3 WHEN, WHAT & HOW MUCH IS PRETRAINING KNOWLEDGE FORGOTTEN?

In this section, we ask three questions:

1. *When is pretraining knowledge forgotten?*
   Our analysis spans four widely used continual-training regimes: (i) domain-continual training (§3.1), (ii) instruction tuning (§3.2), (iii) light SFT/RL on reasoning traces, and (iv) large-scale SFT/RL for reasoning (§3.3). In total, we evaluate almost 30 model–training combinations reflecting common post-training practice, providing broad coverage of how contemporary LLMs are post-trained in the wild. Each post-trained model is compared with its initial checkpoint (§I).

2. *What pretraining knowledge is forgotten?*
   We evaluate each model on 12 public benchmarks, collectively subdivided into close to 100 total subdomains. To summarize systematic patterns, we cluster sub-benchmarks into nine semantically coherent groups that exhibit similar forgetting trends (e.g., commonsense, culture, logic, language, liberal arts, science/tech). These clusters provide a better map of which pretraining knowledge areas are most affected by a given post-training recipe.

3. *How much pretraining knowledge is forgotten?*
   Unless stated otherwise, chance-adjusted metrics for forgetting ($F_{\text{true}}$) and backward transfer ($BT_{\text{true}}$) are used to quantify the severity.

**Experimental setup.** We standardize settings across models for fair comparison. All experiments use the `LightEval` framework (Habib et al., 2023) and log per-sample accuracy. We apply a zero-shot chain-of-thought prompt to instruction-tuned models; base models receive a few-shot prompt solely to teach the format. When available, we add chat-specific templates to be in line with best practices. We cap sequence length at 32,768 tokens[1], except for Qwen2.5-7B-Math and Qwen2.5-7B-Math-Instruct (Yang et al., 2024b), which are limited to 4096. Because base models sometimes continue into subsequent questions, we set explicit stop sequences to end generation once a prediction is produced. Decoding uses temperature $0.6$ with nucleus sampling (`top_p`) of $0.95$. We provide additional details in the Appendix and extensive quantitative results in §I; the following sections present figures and qualitative commentary, with moderate forgetting defined as $15 \pm 5\%$, low as below that threshold, and high as above it. To facilitate reproducibility and further inquiry, we release per-sample logs for every sub-benchmark alongside code.

We now present our results in the subsections below.

## 3.1 SUBAREA 1: DOMAIN-CONTINUAL PRETRAINING

**Motivation.** A popular class of continual learning works adapt general LLMs at the application layer for domains such as coding, mathematics, search, and tool use (Schick et al., 2023; Zhihong Shao, 2024; Ma et al., 2023). As generalist LLMs are increasingly wrapped with tools and domain-specific interfaces, specialization must not erode broad pretraining knowledge. Models still need to contextualize domain outputs, communicate with diverse users, respect cultural norms (Pawar et al., 2025), and uphold safety and ethical standards (Weidinger et al., 2021). These needs motivate our study of forgetting and backward transfer under domain-continual pretraining.

**Setup.** We study continual pretraining that converts a general base model into a specialized one, exemplified by Qwen2.5-Coder (Hui et al., 2024) and Qwen2.5-Math (Yang et al., 2024b).[2] Unlike general instruction tuning or reasoning post-training, domain-continual pretraining shifts the underlying representation using large, relatively uncurated, web-scale domain corpora.

**Main results.** Figure 3 summarizes our findings. Domain-continual pretraining induces little to moderate amounts of forgetting among all post-training methods we evaluate. Backward transfer to general abilities is weak: Gains in the specialized domain rarely improve non-target tasks. The effect spans categories of pretraining knowledge, with no single category driving it, although math-specialized models show significantly more forgetting. Lastly, larger models forget less and have marginally better backward transfer.

---

[1]This budget was sufficient in practice; we never required more tokens.

[2]We treat domain-continual reasoning via SFT/RL separately in §3.3 and focus on domain-continual training here.

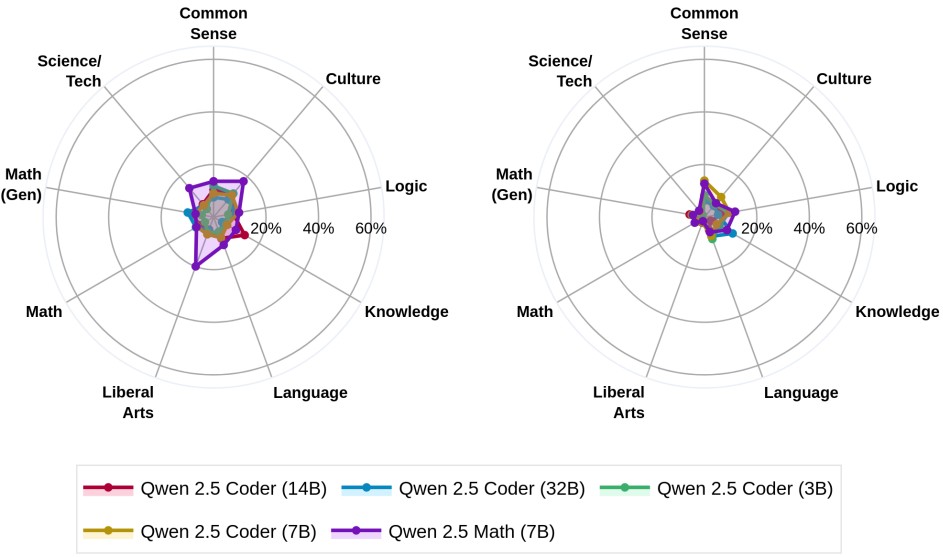

Figure 3: **Forgetting (left) and Backward Transfer (right) after domain-continual pretraining.** Forgetting is low-to-moderate and consistent across categories; backward transfer is low. Scaling model size reduces forgetting.

**Qualitative analysis.** We performed manual error analysis, which indicates reduced instruction-following fidelity (e.g., weaker adherence to constraints, formats, and role-specific directives). Evidence of this is found in supplemental tests using zero-shot chat-template evaluation. In this case, a coder model may, for example, answer "Who was the president of the US?" with a response followed by code, often with embedded answers, making extraction of the true answer difficult. While few-shot prompting alleviates this, it demonstrates a weakened instruction-following ability and less easily elicited knowledge.

> **Takeaway**
>
> Domain-continual pretraining yields low-to-moderate forgetting across categories; backward transfer is limited. Scaling model size marginally reduces forgetting. This suggests that current domain-continual pretraining pipelines alleviate much of the large forgetting behavior seen in previous literature.

## 3.2 SUBAREA 2: INSTRUCTION TUNING

**Motivation.** Base models often require carefully engineered prompts to elicit pretraining knowledge, limiting usability (Brown et al., 2020; Min et al., 2022). Modern post-training pipelines therefore add instruction tuning to enable natural user interaction with minimal prompting (Ouyang et al., 2022; Wei et al., 2021). Most continual-learning work we surveyed focuses on mitigating forgetting in this setting (Wu et al., 2024). We ask: To what extent does instruction following come at the expense of previously learned knowledge?

**Setup.** We measure forgetting and backward transfer from instruction tuning in generalist models, Qwen2.5 (Yang et al., 2024a) and Llama 3.1 (Dubey et al., 2024), as well as domain-continual pretrained models, Qwen2.5-Coder (Hui et al., 2024)[3].

**Results.** As shown in Figure 4, there is low to moderate forgetting across models, with spikes in the Culture and Knowledge categories. However, there is substantial backward transfer in the Math category. Furthermore, scaling model size reduces forgetting and increases backward transfer. This effect is consistent across domain-general and domain-specific base models. While most of the continual learning literature focuses on reducing forgetting in this area, we note that forgetting is low to moderate with current training practices.

---

[3]Qwen2.5-Math Instruct is notably tuned with GRPO, leading us to classify it under Reasoning.

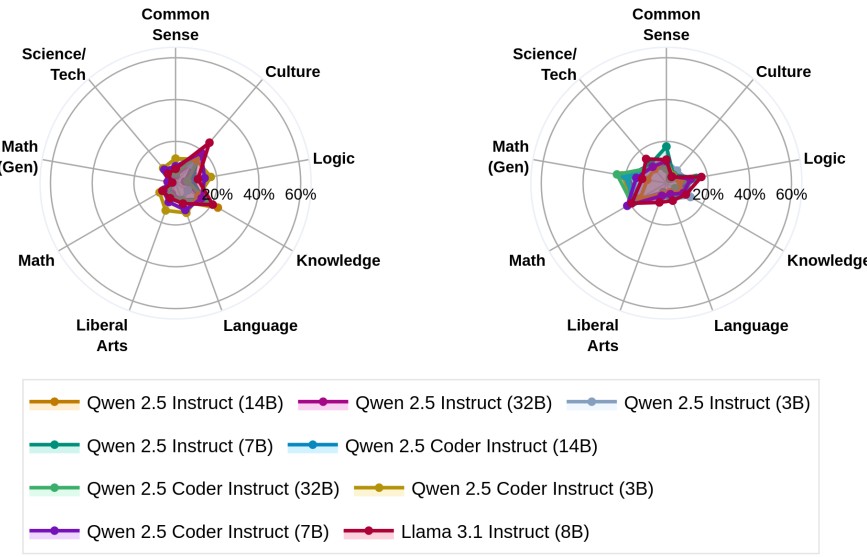

Figure 4: **Forgetting (left) and Backward Transfer (right) after instruction-tuning.** Instruction tuning yields moderate forgetting and backward transfer across categories; scaling model size reduces both.

**Qualitative analysis.** Transfer gains likely reflect better elicitation of pretraining knowledge: instruction-tuned models can leverage what they already know using the straightforward prompts, whereas base models often require carefully crafted prompts.

> **Takeaway**
>
> Instruction tuning produces low-to-moderate forgetting overall and moderate backward-transfer, particularly in math, across model families; the forgetting and backward transfer tend to decrease with increasing model scale. Shifting focus to other subareas of post-training might spur interesting research directions, but there is still progress to be made in this area.

### 3.3 SUBAREA 3: TRAINING WITH REASONING TRACES (SFT AND RL)

**Motivation.** Recent methods encourage explicit reasoning by letting models *think* on a scratchpad before answering (Wei et al., 2022; Kojima et al., 2022), which are now scaled in size and trace length with RL objectives (DeepSeek-AI, 2025). As training domains and data grow, we measure how much such reasoning training induces forgetting to guide continual-learning practice.

**Setup.** We consider two settings: (i) starting from a base model and (ii) starting from an instruction-tuned model. For the latter, we separate light-touch post-training (small datasets) from heavy post-training. We do not separate RL from SFT as the behaviors with respect to forgetting and backward transfer are similar across the two objectives.

#### 3.3.1 TRAINING WITH REASONING TRACES FROM BASE MODELS

**Models.** We evaluate QwQ-32B (from Qwen2.5-32B Base) (Qwen Team, 2025), Qwen2.5-Math-7B-Instruct (RL post-trained with GRPO), and DeepSeek-R1-Distill models distilled from different base models (Qwen2.5 Base and Llama 8B Base) (DeepSeek-AI, 2025).

**Results.** From Figure 5, across scales, model families, and training types, we observe large backward-transfer gains, particularly in Math and Logic, with minimal forgetting. Forgetting is generally low, but moderate for Knowledge and high for Culture. The exception to this trend is the Qwen2.5 Math Instruct model which shows substantial forgetting across many categories. Sample-wise inspection shows that this is primarily due to weak adherence to the prompt, sometimes outputting random

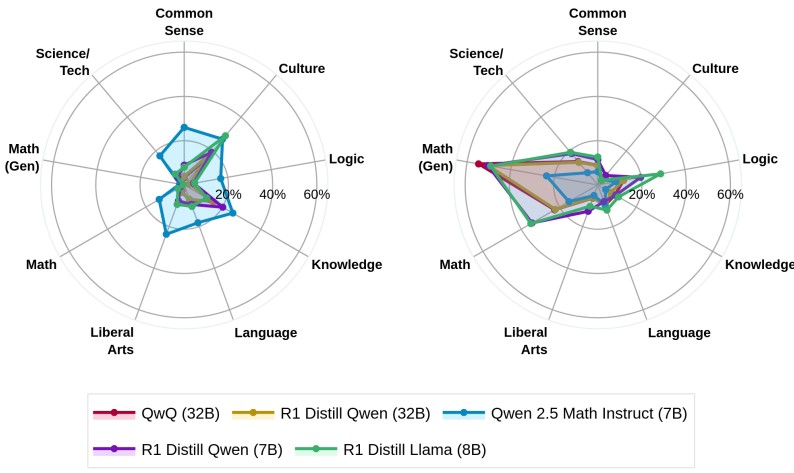

Figure 5: **Forgetting (left) and Backward Transfer (right) after reasoning training (SFT/RL) from base model.** It generally yields minimal forgetting, except in the Culture and Knowledge categories, and has moderate to high backward-transfer gains. Qwen2.5 Math Instruct (7B) is an exception to this trend, demonstrating forgetting across all categories.

multilingual text. With this exception aside, compared to instruction tuning on the same base model (Figure 4), we see similar forgetting and larger backward transfer [4].

We conclude that much of the backward transfer reflects improved instruction following. To isolate reasoning effects beyond elicitation, the next sections analyze reasoning training that starts from an instruction-tuned model. However, models with light-touch reasoning training (i.e. low data) behave differently from those trained at scale (i.e. high data). We therefore present these two cases separately.

> **Takeaway**
>
> Training with SFT/RL for reasoning results in dynamics similar to instruction tuning, but to an even greater extent: We generally observe low to moderate forgetting overall and larger category-specific backward transfer gains. Forgetting mitigation in this domain should consider broad categories of knowledge and abilities when measuring forgetting and backward transfer.

### 3.3.2 REASONING TRAINING FROM INSTRUCTION-TUNED MODELS: LOW-DATA SCENARIO

**Models.** We use the s1.1 family (7B, 14B, 32B) (Muennighoff et al., 2025) and LIMO (v1 and v2) (Ye et al., 2025), all tuned from correspondingly sized Qwen instruct models.

**Results.** Figure 20 summarizes our findings. Across categories, models show minimal forgetting and low backward transfer, except in generative math, where large gains occur. This makes sense, as training for a few passes on little data leaves pretraining knowledge largely intact. That is, the model does not forget much, but it also exhibits few backward transfer gains beyond the instruction-tuned baseline. Scaling model size marginally lowers forgetting, and the smaller teacher–student gap similarly tends to reduce backward transfer, with the exception of the Knowledge category.

> **Takeaway**
>
> For the low-data regime, reasoning training from instruct models yields low forgetting and backward transfer. Forgetting decreases with model scale; backward transfer gains also tend to fall with a narrowing student-teacher gap. This suggests that future forgetting mitigation literature on reasoning models should focus on medium-to-large sized training datasets.

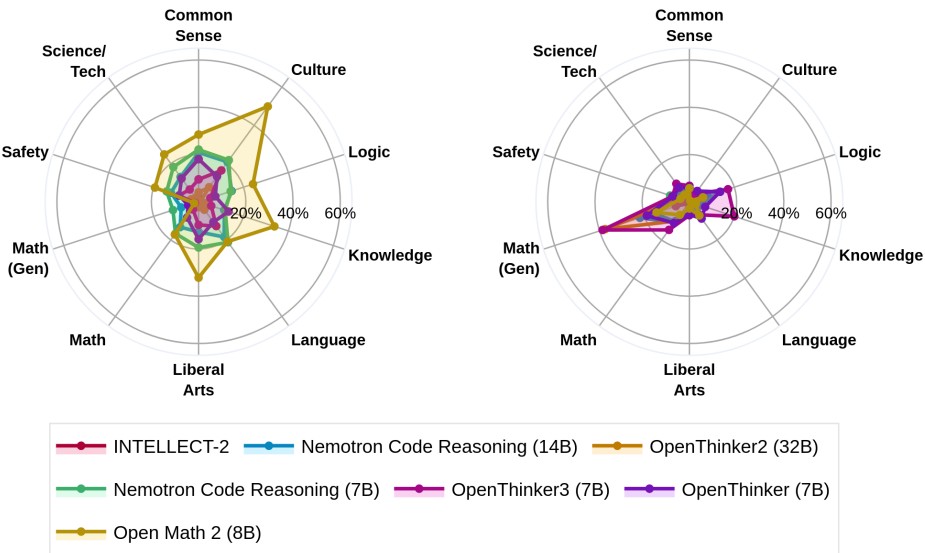

Figure 6: **Forgetting (left) and Backward Transfer (right) after reasoning training from instruct: high data scenario.** No single factor robustly explains the dynamics of forgetting and backward transfer.

### 3.3.3 Reasoning Training from Instruction-Tuned Models: High-Data Scenario

**Models.** We evaluate OpenCodeReasoner (Ahmad et al., 2025), OpenMath2 (Toshniwal et al., 2024), OpenThinker-7B, OpenThinker2-32B, and OpenThinker3-7B (Guha et al., 2025), and Intellect-2-32B (Prime Intellect Team et al., 2025). This spans SFT (former) and RL (Intellect-2).

**Results.** Results vary by domain mix and model quality. The OpenThinker models generally show low–to–moderate forgetting and moderate backward transfer, perhaps due to the breadth of their training data mix, whereas OpenCodeReasoner models show consistently high forgetting with low backward transfer gains due to their narrower training data. Furthermore, we find this may be primarily due to weakened instruction-following capabilities, as sample-level inspection shows the model will refuse to answer with letters when numbers are present as options, instead answering numerically. This is also seen with the Nemotron Code Reasoning models, where answers will often be embedded within python code. These factors can make the forgetting and backward transfer observed highly dependent on the extraction method used. We account for this through LLM-as-a-judge evaluation (§ C.4). Scaling model size, as seen in the OpenThinker models, signals improvements in both forgetting and backward transfer — consistent with most previous sections. Decentralized training (as in Intellect-2), in contrast, showed minimal forgetting or backward transfer. We conjecture that the model largely remains unchanged given that it shows negligible gains on the optimized math benchmarks Hochlehnert et al. (2025). However, the results here remain preliminary. We do not find a single dominant factor, whether initialization, data regime, or scale, that sufficiently explains forgetting and backward-transfer dynamics. We believe controlling the finer details that determine the quality of the trained model might lead to better conclusions.

> **Takeaway**
>
> No single factor robustly explains the dynamics of forgetting and backward transfer; training on a mix of domains appears to reduce forgetting and improve backward transfer.

## 4 Does Model Merging Reduce Forgetting?

**Motivation.** Recent work shows that offline model merging can combine capabilities from multiple models (Dziadzio et al., 2025). Unlike classical continual learning (De Lange et al., 2022), model merging requires neither the original training data nor the ability to resume training, which is practical in resource-constrained settings.

---

[4]All corresponding tables are available in §I.3 for detailed comparison.

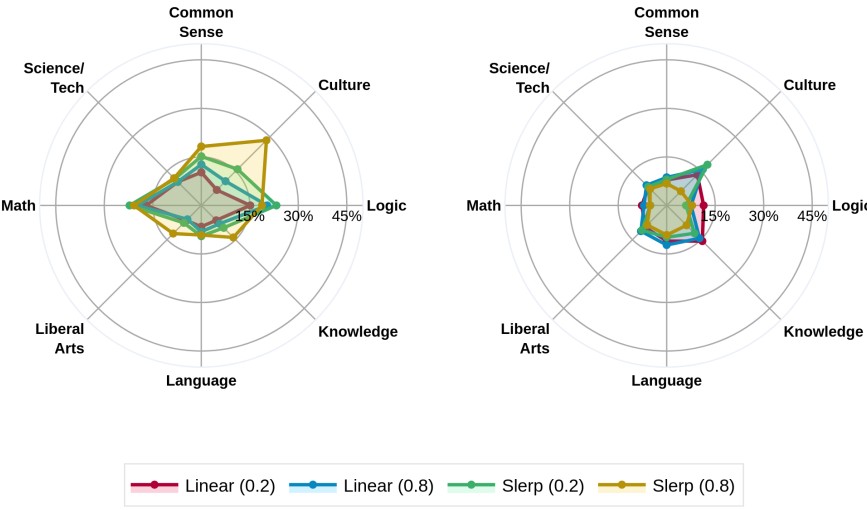

Figure 7: **Forgetting (left) and Backward Transfer (right) of Qwen 2.5 Base merged with Qwen 2.5 Coder (7B) relative to Qwen2.5 Coder.** The merged model induces moderate forgetting and little backward transfer.

**Setup.** We evaluate Exponential Moving Average (EMA) merging; in the two-checkpoint case this is linear interpolation,

$$\theta_{\text{EMA}}(\alpha) = \alpha \, \theta_{\text{pre}} + (1 - \alpha) \, \theta_{\text{post}}.$$

Prior large-scale studies find these simple schemes effective for continual learning with foundation models (Roth et al., 2024). Our experiments compare linear interpolations (e.g., LERP and SLERP) across OpenThinker-7B, OpenThinker3-7B, and Qwen2.5-Coder-7B, together with their base checkpoints.

**Results.** We compare merged checkpoints to the post-trained model $\theta_{\text{post}}$; results for $\theta_{\text{pre}}$ appear in the Appendix. For Qwen2.5-Coder-7B and OpenThinker3-7B, even small mixes with the base checkpoint degrade performance, severely so for the latter (Figures 7, 24). In contrast, OpenThinker-7B shows small overall gains, accompanied by moderate forgetting (Figure 28). In our setting, merging does not mitigate forgetting. This may reflect that we merge only two checkpoints, whereas prior work often merges eight or more (Yadav et al., 2023; 2024). We further hypothesize that weight drift between our checkpoints is larger than typical for models in the merging literature, which could explain these outcomes.

> **Takeaway**
>
> Merging models does not yet reliably mitigate forgetting in post-training pipelines.

Merging remains promising, but further study is needed to how to overcome its limitations and whether increased scale can compensate for these difficulties.

## 5 CONCLUSION

We present a new metric for sample-wise forgetting and backward transfer that corrects for chance in multiple-choice evaluations. Our results challenge the claim that sequential training induces catastrophic forgetting of pretraining knowledge, showing this is not universally true. Forgetting depends on the post-training method and its scale. By focusing on sample-wise forgetting, we offer a clearer map of what knowledge is lost and at what stages of post-training language models lose knowledge, providing fertile ground to study how to preserve and accumulate knowledge and capabilities by post-training. Promising ways to prevent forgetting include: (1) Designing objectives and data that explicitly penalize 1→0 transitions; (2) Using targeted synthetic corpora or brief mid-training bursts to repair localized forgetting; (3) Adding retrieval mechanisms to reduce reliance on in-weight knowledge storage.

ACKNOWLEDGMENTS

AP and MB acknowledge financial support by Federal Ministry of Research, Technology and Space (BMFTR) FKZ: 16IS24085B and Open Philanthropy Foundation funded by the Good Ventures Foundation. AH acknowledges funding by the Federal Ministry of Research, Technology and Space (BMFTR), FKZ: 16IS24079A. AH thanks the International Max Planck Research School for Intelligent Systems (IMPRS-IS) for support.

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

# Appendix

CONTENTS

## A  RELATED WORK

**Post-training techniques.**  A broad set of post-training methods now underpins standard LLM pipelines. *Supervised fine-tuning (SFT)* (Ouyang et al., 2022) remains the core step, used for continued pretraining and instruction tuning. At later stages, *reinforcement learning from human feedback (RLHF)* (Ouyang et al., 2022) aligns model outputs with human preferences. To simplify preference learning, *direct preference optimization (DPO)* (Rafailov et al., 2023) provides a direct loss surrogate. With the rise of test-time scaling (e.g., sampling depth or compute at inference), *group relative policy optimization (GRPO)* (Shao et al., 2024) has been proposed to elicit stronger intrinsic reasoning. Taken together, these methods introduce distinct objectives and optimizers, increasing the complexity of the post-training stack (Wang et al., 2025).

**Measuring catastrophic forgetting.**  Catastrophic forgetting is the loss of previously acquired knowledge when a network learns new information. Early studies examined the effect in small models and simplified settings (McCloskey & Cohen, 1989; Ratcliff, 1990; French, 1999). Lopez-Paz & Ranzato (2017) formalized forgetting via *backward transfer*, the effect of learning a new task on performance in earlier ones: positive values indicate improvement; negative values indicate forgetting. Recent work extends these analyses to deep networks trained on large-scale data, with growing attention to language models (Biesialska et al., 2020; Wu et al., 2022).

**Benchmark paradigm.**  Task-incremental learning is the dominant paradigm for benchmarking forgetting (De Lange et al., 2022). Models learn a sequence of tasks with clear boundaries, and task labels are available at train and test time. Class-incremental learning removes test-time task identifiers, making evaluation stricter (Wang et al., 2024). Other views analyze continual learning through positive/negative transfer (Yıldız et al., 2025). At the sample level, Toneva et al. (2019) introduced forgetting metrics that identify "unforgettable" examples (stable once learned) and "catastrophically forgotten" examples (highly plastic), and showed these patterns are consistent across architectures and random seeds.

**Language-model forgetting.**  Recent studies focus on forgetting induced by instruction tuning. Luo et al. (2025) trained models of up to 7B parameters with SFT and evaluated multiple knowledge categories. DeepSeek-AI (2024) reported instruction-tuning-related regressions on sentence completion even for 67B models. Fernando et al. (2025) examined forgetting across SFT followed by RLHF and proposed joint-training strategies to mitigate it. Lin et al. (2024) framed instruction-tuning degradation as an "alignment tax" (performance loss on pretraining skills due to alignment) and found model merging to be the most effective mitigation under a Pareto-efficient trade-off among tested techniques. Li & Lee (2024b) studied continual pretraining on aligned LMs and observed notable regressions in alignment-related behavior.

**Catastrophic forgetting in reasoning training pipelines.**  Work on reasoning-oriented LMs highlights new failure modes. Li et al. (2025) defined *temporal forgetting*: models lose the ability to solve problems they could solve at earlier training checkpoints. The effect appears in both RL-trained and instruction-tuned models. They proposed *temporal sampling*—round-robin sampling from recent checkpoints—as a mitigation. Pipatanakul et al. (2025) merged a language-fine-tuned model with DeepSeek R1 Distill (70B; both derived from Llama 3.3 70B (Dubey et al., 2024)) to adapt reasoning while preserving language competence. For multimodal models, Chen et al. (2025) found that later layers primarily support reasoning, whereas early layers handle perception, suggesting layer-wise interventions. We extensively document forgetting across post-training pipelines.

**Mitigation strategies.**  Sequential SFT to RLHF/DPO can exacerbate forgetting. To counteract this, researchers explore: (i) *model averaging*, interpolating between pre- and post-RLHF checkpoints to trade off alignment and retention (Lin et al., 2024); (ii) *joint post-training*, optimizing supervised and preference objectives simultaneously with convergence guarantees (Fernando et al., 2024); and (iii) *unified fine-tuning (UFT)*, which folds instruction tuning and alignment into a single implicit-reward objective (Wang et al., 2025). Additional techniques—including advantage models and selective rehearsal—stabilize RLHF by shaping reward distributions and replaying curated data (Peng et al., 2023). *Online Merging Optimizers (OMO)* combine gradients from SFT and RLHF models during training to maximize reward while preserving pre-trained skills (Lu et al., 2024). Theory supports these interventions: up to permutation symmetries, weights of homologous models tend to lie in a

shared low-loss basin (Ainsworth et al., 2023). Hence, we were quite surprised that model merging does not work for our simple case of mitigating forgetting during post-training with only two deep networks.

**Forgetting at scale.** Pretraining mitigates forgetting relative to training from scratch (Mehta et al., 2023; McRae & Hetherington, 1993). Ramasesh et al. (2022) further found that pretrained ResNets and Transformers (up to ∼100M parameters) are robust to forgetting at scale; language experiments showed similar trends. However, Luo et al. (2025) reported increased forgetting with scale in the 1–7B LM regime, suggesting modality- and regime-dependent behavior. In contrast to these works, we study forgetting during post-training of language models.

# B  EXPERIMENTAL SETUP

## B.1  EVALUATION

We standardize settings across models for fair comparison. All experiments use the `LightEval` framework (Habib et al., 2023) and save sample-wise results, such as accuracies. We apply a zero-shot chain-of-thought prompt to instruction-tuned models and require answers in a fixed MCQ format; base models receive a few-shot prompt solely to teach the format. When available, we add chat-specific templates to be in line with best practices. We cap sequence length at 32,768 tokens, except for Qwen2.5-7B-Math and Qwen2.5-7B-Math-Instruct (Yang et al., 2024b), which are limited to 4,096 tokens. Decoding uses temperature 0.6 with nucleus sampling (`top_p`) of 0.95. All datasets are evaluated on at least 3 seeds and the metrics are reported with mean and standard deviation (§I). To facilitate reproducibility and further inquiry, we release per-sample logs for every sub-benchmark alongside code.

Chat-specific templates are incorporated into the prompt to ensure consistent formatting and fair evaluation. Because some models, particularly base models, tend to continue generating responses for subsequent questions after completing the current one, we provide explicit stop sequences to terminate generation once a prediction has been produced. For the case of base models, few-shot prompting yields a more accurate elicitation of their knowledge, as otherwise the model can output the answer in undesired formats (§G.2).

To encourage strict MCQA formatted answers for instruction-tuned models we prepend an instruction prompt with explicit answer formatting guidelines with an example (§G.1). This is necessary for accurate evaluation since models sometimes output the option text or phrases instead of the letter. In particular, we find including additional instructions to not use extra punctuation, asterisks, lowercase letters for answers, or trailing spaces necessary because LightEval's default regex and letter extractors can fail in certain cases otherwise. We additionally tell the model to constrain its output to be of the form "Answer: $LETTER" as otherwise models will often provide the corresponding answer to a given letter or provide the answer in another format, making extraction more prone to error.

For datasets where CoT reasoning traces are provided for few-shot prompting, we use those. For free-form/generative math questions, we follow the prompt and extraction methods used in Hochlehnert et al. (2025).

## B.2  DATASETS

To evaluate broad model knowledge and capabilities we benchmark on eighteen public datasets: MMLU (Hendrycks et al., 2021b;a), BBH (Suzgun et al., 2023), GPQA (Rein et al., 2024), MuSR (Sprague et al., 2024), ARC (Clark et al., 2018), TruthfulQA (Lin et al., 2022), HellaSwag (Zellers et al., 2019), Social IQa (Sap et al., 2019), MCTest (Richardson et al., 2013), PIQA (Bisk et al., 2020), CommonsenseQA (Talmor et al., 2019), SaladBench (Li et al., 2024), AIME24 (AI-MO, 2024), AIME25 (Yen-Ting, 2025), AMC23 (Engineering, 2025a), Math500 (HuggingFaceH4, 2024), Minerva (Engineering, 2025b), and OlympiadBench (He et al., 2024). Several of these benchmarks, namely MMLU and BBH, provide subcategory labels which allow for splitting into further sub-benchmark evaluations by subject. To enable easier understanding, we group these (sub-)benchmarks into high-level groups used to evaluate the capabilities of the models. They are grouped such that (sub-)benchmarks in the same group show similar trends in forgetting and backward transfer (§H).

## C   METHODOLOGICAL ANALYSIS

Evaluating a wide range of models and datasets requires careful attention to the form of prompting, metric comparison, metric robustness, and extraction accuracy. We check each of these categories to verify assumptions and determine the way to conduct experiments that most accurately measures true the change of knowledge in models under post-training.

### C.1   PROMPTING

When evaluating base models, prompting, in particular, becomes an important factor to consider in assessing the correctness. Therefore, we first measure the ability of base models using the same prompting as instruction-tuned models. Under these conditions, we see ostensibly large forgetting in domain-continual pretrained models (Figure 8). Our qualitative analysis suggests that this is largely due to the models output in an incorrect format, such as in code, where the location of the answer can be obscured. When contrasted with few-shot prompting, where there is much less forgetting, we conclude that forgetting metrics can vary significantly depending on how knowledge is elicited, especially when training on narrow tasks, an effect that few-shot prompting alleviates. Therefore, in all experiments, unless otherwise stated, we evaluate base models with few-shot prompting, as it elicits the learned knowledge better. For these reasons, measuring the performance of base

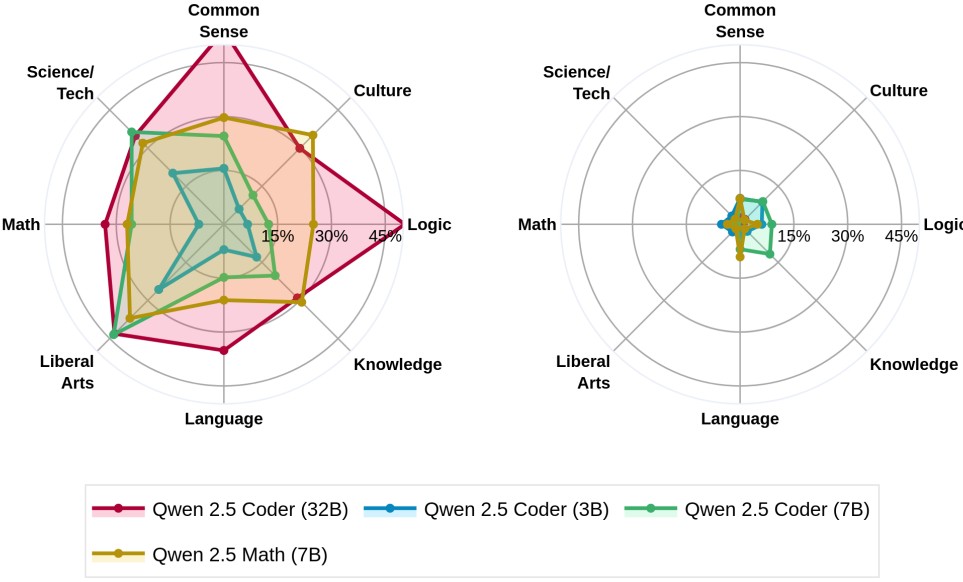

Figure 8: **Forgetting (left) and Backward Transfer (right) of Domain-Adaptive Pretraining models evaluated with zero-shot chat template prompting instead of few-shot prompting.** Substantial specious forgetting is observed relative to Figure 3, motivating our use of few-shot prompting for all base model evaluations.

models in behavioral evaluations can become nontrivial. While benchmarks measuring knowledge or capabilities may be elicited through a few-shot prompt, others, such as truthfulness or safety, become more difficult because prompting them with examples would bias their behavior (Lin et al., 2022). Future work should consider the effect of no-knowledge few-shot prompting, where the question-and-answer format is demonstrated without leaking content examples, to avoid biasing the base model's output.

> **Evaluation Decisions**
>
> Base models are evaluated with few-shot prompting. Safety evaluations are only conducted when comparing two instruction-tuned models.

### C.2 SAMPLE VS. AGGREGATE METRIC COMPARISON

Another key consideration is empirical evidence demonstrating that sample-wise metrics uncover more forgetting and backward transfer than aggregate metrics. A theoretical argument in favor of sample-wise metrics over the standard aggregate metric, defined as

$$F_{\text{standard}} = \max(\bar{a}^{\text{pre}} - \bar{a}^{\text{post}}, 0)$$

has indicated this to be possible, but it is unclear a priori the degree to which this occurs in practice. We confirm this to play a large role in uncovering forgetting in Figure 9. Our sample-based metric reveals substantially more forgetting relative to the standard formulation, in some cases finding what was originally low forgetting is actually moderate. This highlights sample-level degradation that is otherwise hidden when averaging over tasks. A practitioner observing $F_{\text{standard}} \approx 0$ might reasonably conclude that post-training preserved prior capabilities intact, yet this apparent stability can mask substantial changes. Simultaneous forgetting and backward transfer that cancel in expectation but reflect a fundamentally different model. In domains such as legal or medical reasoning, where practitioners may rely on a model's established behavior for specific query types, such hidden redistribution of competence poses real deployment risks that aggregate metrics fail to flag.

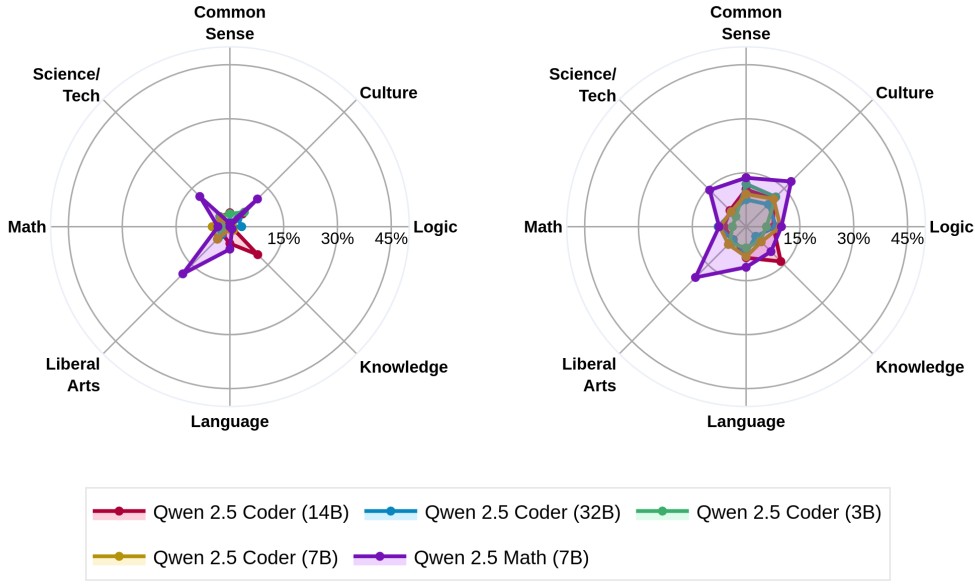

Figure 9: **Coder model comparing conventional forgetting (left) against our sample-wise forgetting (right).** More (genuine) forgetting is uncovered when using the sample-wise forgetting metric.

> **Evaluation Decisions**
>
> Sample-wise metrics are used over standard aggregate metrics.

## C.3 Metric Robustness under MCQA

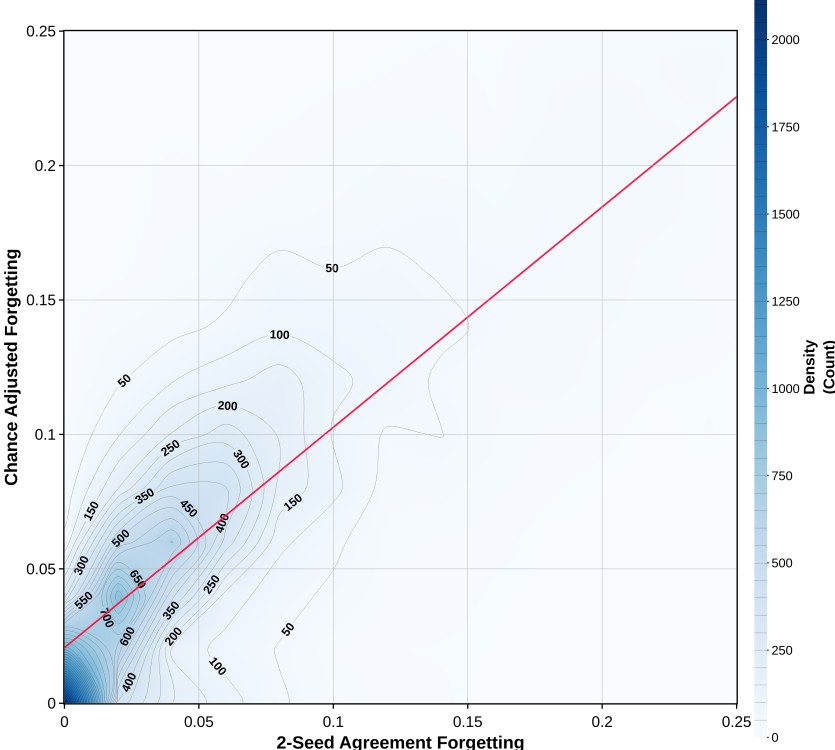

Figure 10: **Sub-benchmark count across 2-Seed Agreement Forgetting (x-axis) and Chance-Adjusted Forgetting (y-axis).** The line of best fit (red) shows both metrics are highly correlated across sub-benchmarks.

We review the robustness of the chance adjusted forgetting in measuring true knowledge loss, which is particularly relevant when evaluating under MCQA benchmarks as models can often guess the answer correctly. While our metric accounts for this by subtracting out an estimate of this probability, we compare this to another sample-level metric which filters out noisy samples to empirically demonstrate this. Namely, we consider samples where there is agreement among two out of three seeds on average, and consider the cases where there is forgetting relative to the other cases.

$$F = \frac{(1 \to 0)_2}{(0 \to 0)_2 + (0 \to 1)_2 + (1 \to 0)_2 + (1 \to 1)_2}.$$

where we formally define the two-seed sample-agreement metric as follows:

$$(x \to y)_2 := \frac{1}{\binom{|\mathcal{S}|}{2}} \sum_{\{s,t\} \in \binom{\mathcal{S}}{2}} \sum_{i=1}^{N} \mathbf{1}\left\{a_{i,s}^{\mathrm{pre}} = x = a_{i,t}^{\mathrm{pre}} \ \wedge \ a_{i,s}^{\mathrm{post}} = y = a_{i,t}^{\mathrm{post}}\right\}.$$

where $\mathcal{S}$ is the set of seeds and $N$ the number of samples. Intuitively, this measures robust knowledge loss relative to stable knowledge. We find that this metric agrees with our results across the post-training pipeline, thereby indicating the chance-adjusted forgetting metric captures robust knowledge loss, rather than random forgetting. We show the correlation between these metrics in Figure 10 and provide a direct comparison in Figure 11.

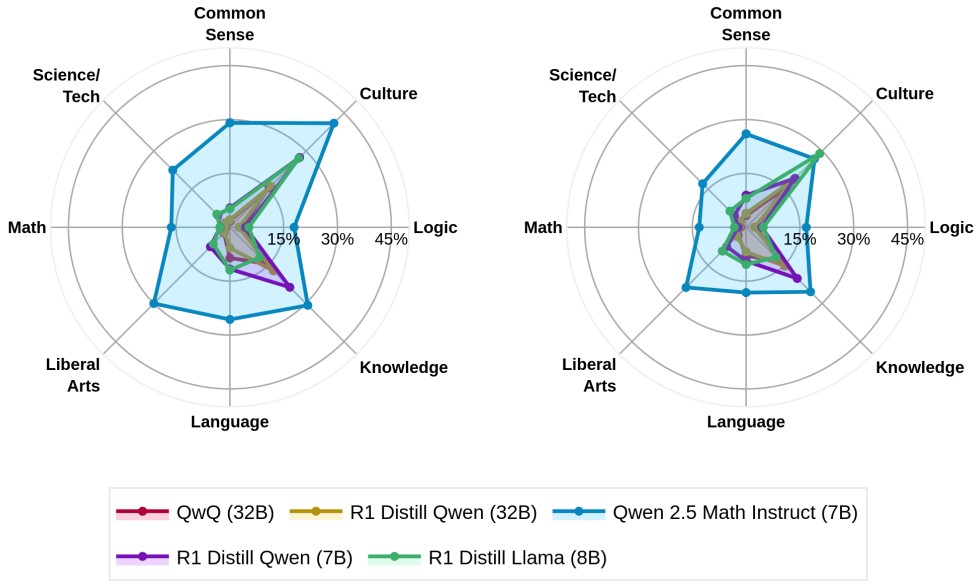

Figure 11: **2 Seed Forgetting (left) and Chance Adjusted Forgetting (right) of Models Trained from Base**. The overall trends remain the same, with only minor differences such as a slight increase in forgetting for Qwen2.5 Math Instruct (7B) in the 2 seed forgetting case.

> **Evaluation Decisions**
>
> Chance-adjusted (sample-wise) metrics are used on MCQA benchmarks.

## C.4 EXTRACTION

Sample-level inspection occasionally shows answers that are correct, but unable to be extracted correctly through the regex extraction. This occurs particularly in models trained for specialized tasks. For example, we find that coding models will assign the correct answer to a variable in code and then provide the answer variable in a print statement. We control for these extraction-related errors by using an LLM extractor, specifically Qwen2.5-14B-Instruct, which we find sufficient to correct for errors. We do this by providing the question, response, and ground truth answers using the prompt in G.3 without a chat template, in order to encourage immediate JSON output.

We find that this primarily corrects for outliers while all trends remain the same, which we find to be true across all knowledge and post-training categories. By comparing Figure 12, where LLM extraction is used, and Figure 13, where regex extraction is used, we see the outlier effect of Qwen2.5 Math Instruct (7B) is reduced. The overall trends remain the same.

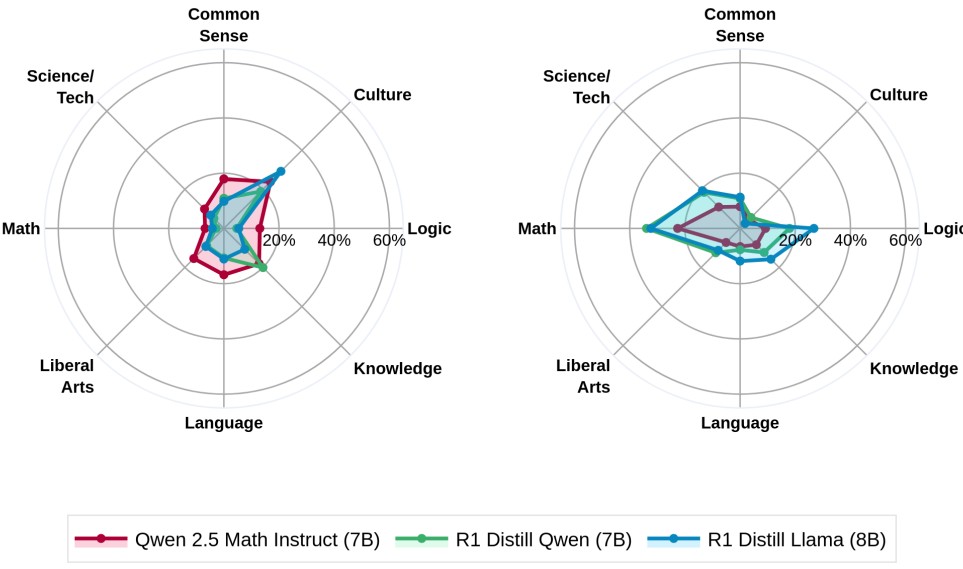

Figure 12: **Forgetting (left) and Backward Transfer (right) of Models Trained from Base using LLM Extraction**. Trends are the same as in Figure 13, but Qwen2.5-Math-Instruct's outlier tendencies are reduced.

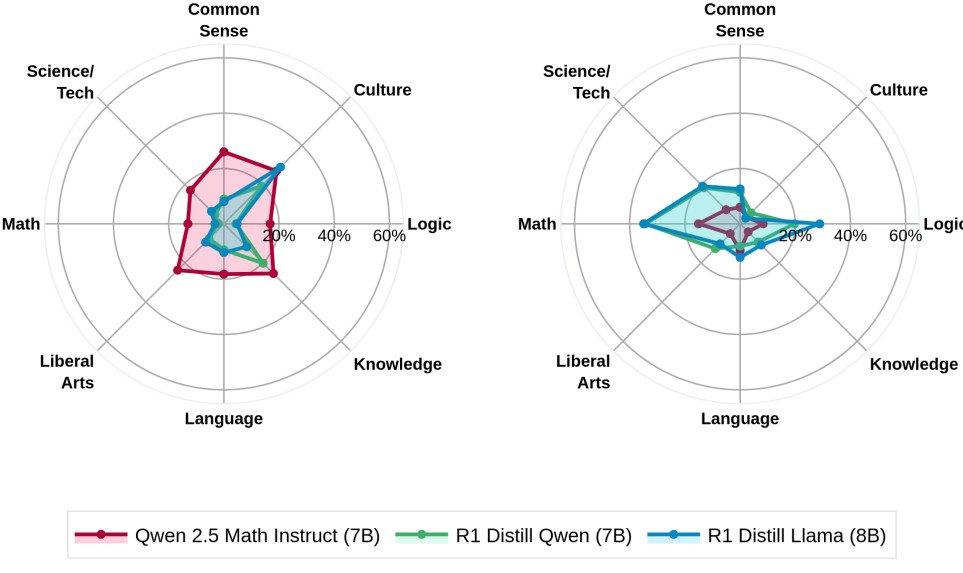

Figure 13: **Forgetting (left) and Backward Transfer (right) of Models Trained from Base using Regex Extraction**. Trends are the same as in Figure 12, but Qwen2.5-Math-Instruct's outlier tendencies are increased.

## D  EXPANDED COMPARISONS

The high-data reasoning training scenario in §3.3.3 encompasses a heterogeneous collection of models that differ along several dimensions simultaneously: data diversity, training objective (SFT vs. RL), and data volume. To isolate the contribution of each factor, this section disentangles these variables by grouping models according to a single axis at a time.

## D.1 DATA DIVERSITY

We split reasoning models into two cases: those trained on narrow domains (one or two benchmark categories), such as math or code, and those trained on mixed data (jointly trained on many tasks or on general data). As indicated in §3.3.3, increased data diversity generally mitigates forgetting and improves backward transfer (Figure 15), whereas decreased data diversity shows the opposite trend (Figure 14). This pattern is consistent with the broader continual-learning intuition that rehearsing a wider range of tasks reduces interference. That is, models trained on mixed data encounter the relevant knowledge distribution more often during training, reducing the probability of overwriting it.

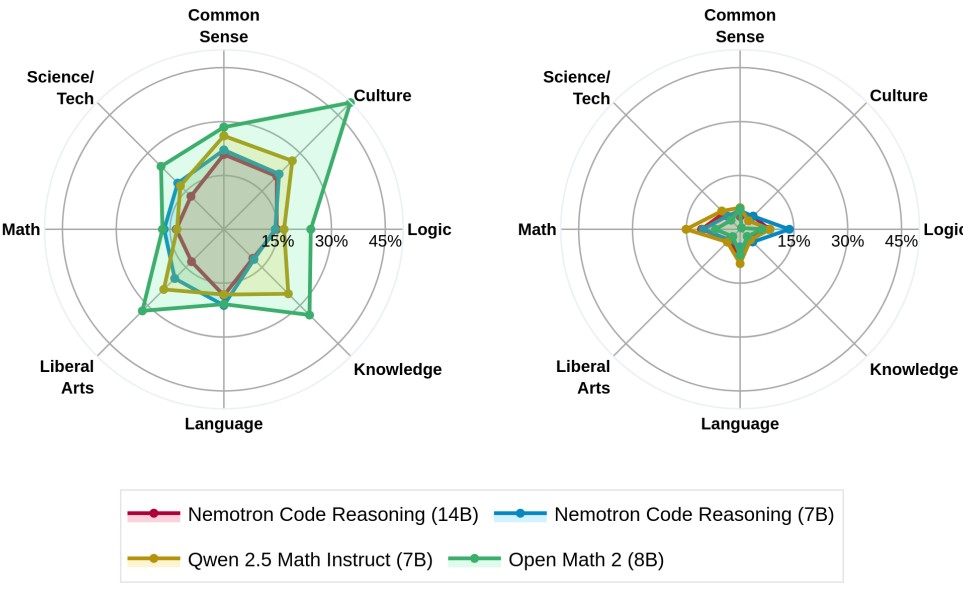

Figure 14: **Forgetting (left) and Backward Transfer (Right) of Reasoning Models Trained with Narrow Data**. Backward transfer is generally low or moderate and forgetting is larger than training on mixed data.

The narrow-data models in Figure 14 demonstrate moderate to high forgetting across most knowledge categories, while backward transfer remains low. This pattern is most pronounced in the Culture, Liberal Arts, and General Knowledge categories, which are structurally furthest from the narrow training distributions of math and code. This is consistent with gradient signals concentrated in a single domain, progressively displacing representations shared with general knowledge (French, 1999). Because narrow-domain training rarely revisits these general-knowledge representations, there is no countervailing signal to preserve them. The result is capability gains in the target domain alongside losses in breadth.

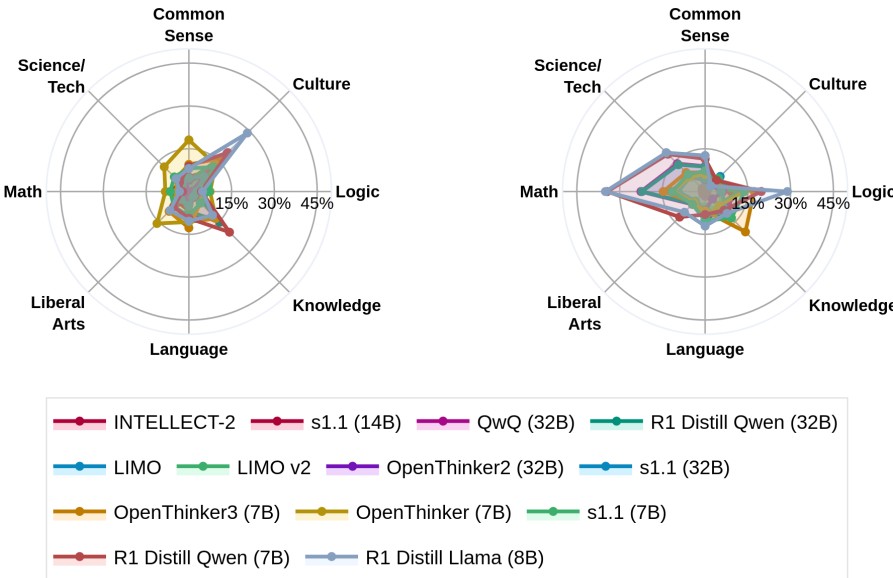

Figure 15: **Forgetting (left) and Backward Transfer (Right) of Reasoning Models Trained with Mixed Data**. Backward transfer is generally moderate to high.

> **Takeaway**
>
> Data diversity tends to mitigate forgetting and improve backward transfer.

## D.2 OBJECTIVE FUNCTION

Reasoning models are trained with a variety of objectives: supervised fine-tuning (SFT) on curated reasoning traces, reinforcement learning (RL) via policy-gradient methods such as GRPO, or a combination of both. In principle, SFT and RL could differ in their forgetting dynamics. SFT directly optimizes a cross-entropy loss over fixed demonstrations, while RL explores a broader policy space and updates are shaped by reward signals that may be more or less correlated with pretraining knowledge. However, in practice, Figures 16 and 17 do not show systematic differences between the two objective types in the models evaluated. Both groups exhibit comparable levels of forgetting and backward transfer, suggesting that the training objective is less important than data composition and volume in determining forgetting behavior.

One potential explanation for this result is that both objectives operate on similar data. For example, an RL policy trained on math problems explores the same knowledge space as an SFT model trained on curated math traces, regardless of whether updates are shaped by cross-entropy or a scalar reward. Therefore, key variable could be the coverage of the training distribution, not the loss function. This interpretation is supported by the heterogeneity within each group; models differ in base checkpoint, data volume, and domain mix yet still converge to comparable forgetting and backward transfer profiles.

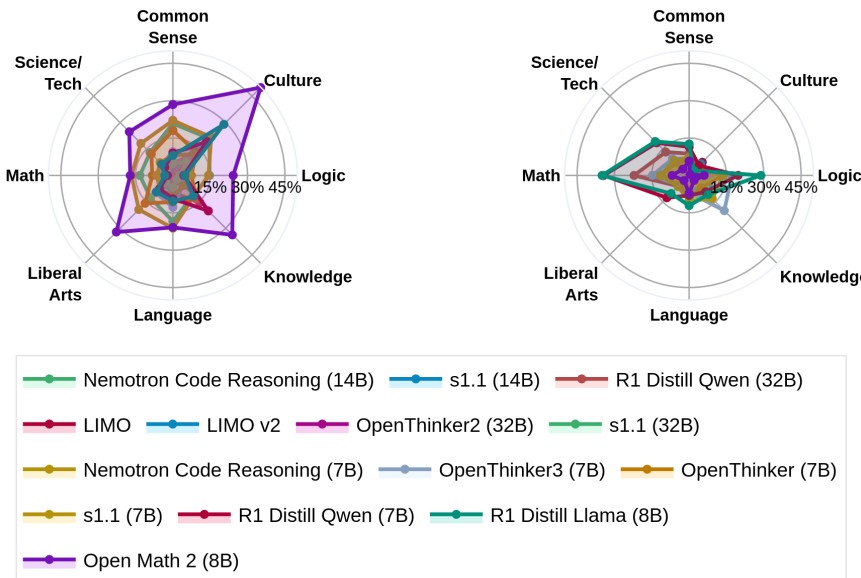

Figure 16: **Forgetting (left) and Backward Transfer (Right) of Reasoning Models Trained with SFT Data**.

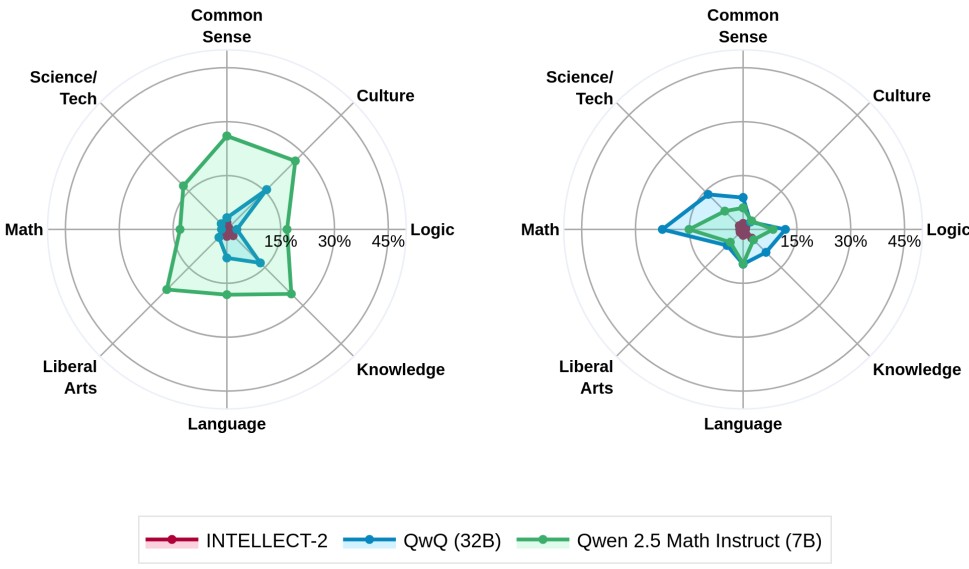

Figure 17: **Forgetting (left) and Backward Transfer (Right) of Reasoning Models Trained with RL Data**.

> **Takeaway**
>
> Differences between SFT and RL reasoning post-training in forgetting and backward transfer dynamics appear to be negligible in practice.

## D.3   DATA VOLUME

Data volume is another important factor, since more training examples give the optimizer more signal to update weights, which could either help, by covering more knowledge, or harm, by overwriting more pre-trained knowledge. Figure 18 shows that models trained on low data volumes exhibit uniformly low forgetting and backward transfer, consistent with the finding in §3.3 that light-touch training leaves pretraining knowledge largely intact. However, for high data volumes (Figure 19), no clear trend emerges. Forgetting and backward transfer vary substantially across models, suggesting that volume per se is not a reliable predictor once data is abundant. At that point, the composition of training data appears to dominate.

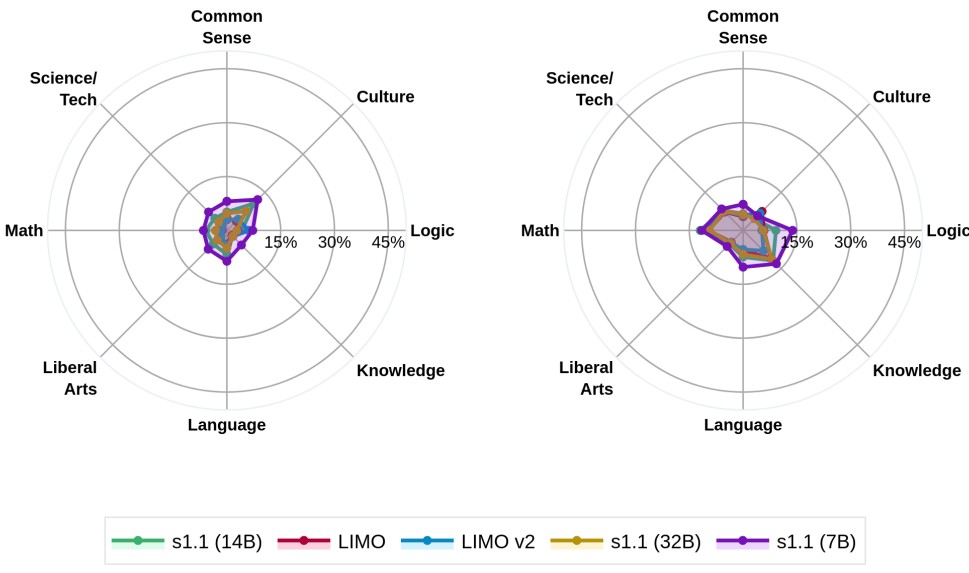

Figure 18: **Forgetting (left) and Backward Transfer (Right) of Reasoning Models Trained with Low Data Volume**. Both metrics are generally low across categories.

The low-volume radar plot in Figure 18 is notably uniform. This uniformity is itself informative. It implies that a carefully-conducted, small training signal is insufficient to meaningfully reshape large categories of pretraining knowledge, regardless of domain. From a continual-learning perspective, low-volume training may occupy a regime where the optimizer takes steps too small to accumulate into large weight displacements, leaving the pretrained basin essentially intact.

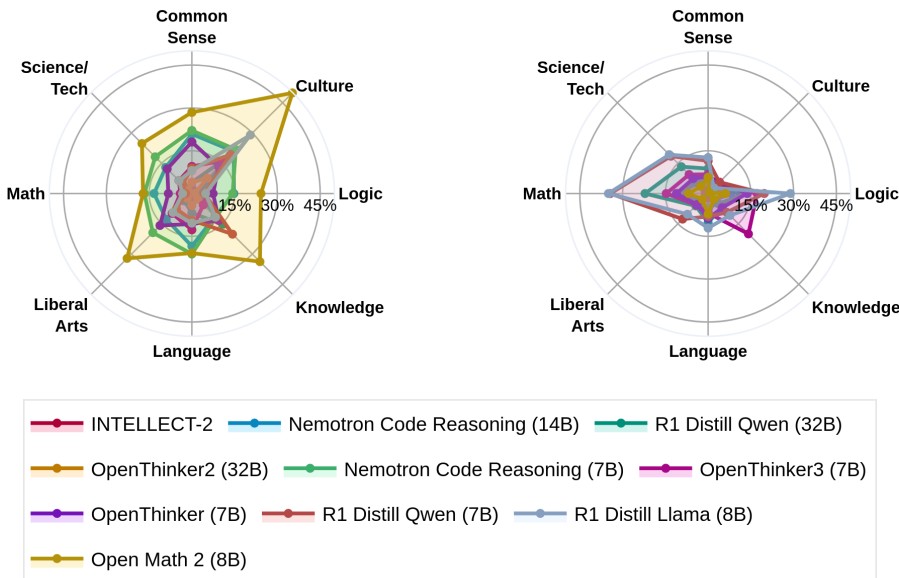

Figure 19: **Forgetting (left) and Backward Transfer (Right) of Reasoning Models Trained with High Data Volume**. Forgetting and backward transfer vary significantly between models.

> **Takeaway**
>
> Low data volume tends to mitigate forgetting.

Taken together, the three ablations in this section point toward data composition as the primary driver of forgetting in reasoning post-training. Data diversity and data volume both show clear monotonic relationships with forgetting, whereas the training objective does not. This ordering has a practical implication. Practitioners seeking to reduce forgetting without sacrificing reasoning gains should first consider the breadth and quantity of their training mix before redesigning the loss function. Specifically, mixing in general-domain data may substantially reduce category-specific forgetting at moderate cost to in-domain performance.

# E SUPPLEMENTARY PLOTS

## E.1 REASONING TRAINING FROM INSTRUCTION: LOW-DATA SCENARIO

Figure 20 shows the full radar charts for the low-data reasoning training scenario described in §3.3. Across all models and categories, both forgetting and backward transfer are low, confirming that light-touch post-training leaves most pretraining knowledge intact. The s1.1 model family (7B, 14B, 32B) shows a consistent decrease in forgetting with scale. Backward transfer gains concentrate almost exclusively in the Math (Generative) category, reflecting the narrow training distribution, with negligible change elsewhere. LIMO models exhibit a similar pattern: backward transfer is confined to generative math and minimal disruption occurs in other knowledge categories.

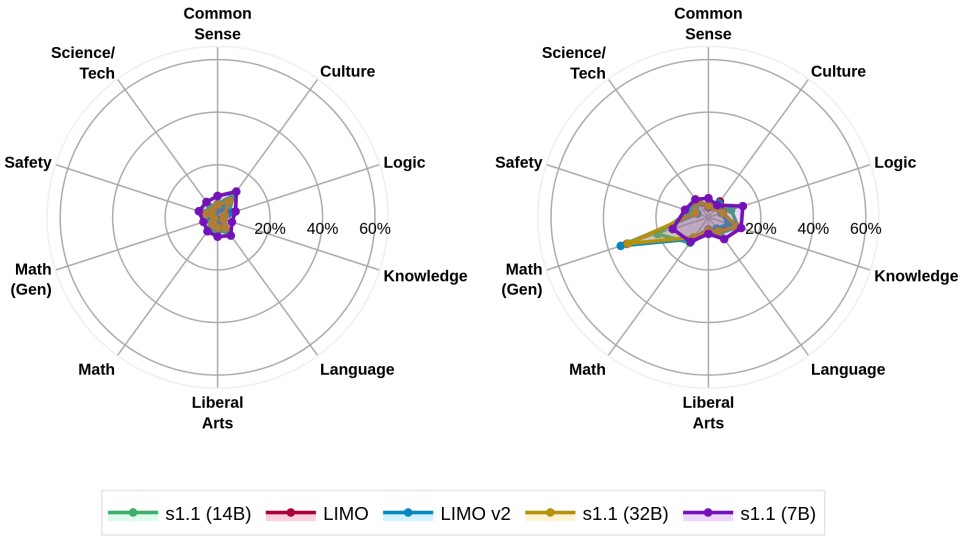

Figure 20: **Forgetting (left) and Backward Transfer (right) after reasoning training from instruct: low data scenario.** Yields little forgetting and backward transfer, with the exception of the Math (Gen) category. Forgetting decreases with model scale.

## E.2 MAXIMUM FORGETTING

We address another potential confounder in our results. Namely, a post-trained model's maximum forgetting (backward transfer) is bounded by its base model (i.e., the model before post-training): if the base model never learned a capability, the post-trained model cannot forget it further. This means that models with stronger base capabilities are inherently exposed to greater forgetting risk, which could confound comparisons across model families. To assess whether this bound is a limiting factor in our analysis, we compare empirical forgetting against the theoretical maximum for each model. Figure 21 and Figure 22 shows that this does not play a significant role in our analysis: in practice, the gap between empirical and maximum forgetting is large across all evaluated models, indicating that post-training does not push models close to their forgetting ceiling. Our conclusions are therefore unlikely to be an artifact of this constraint. Further results are shown in §I.

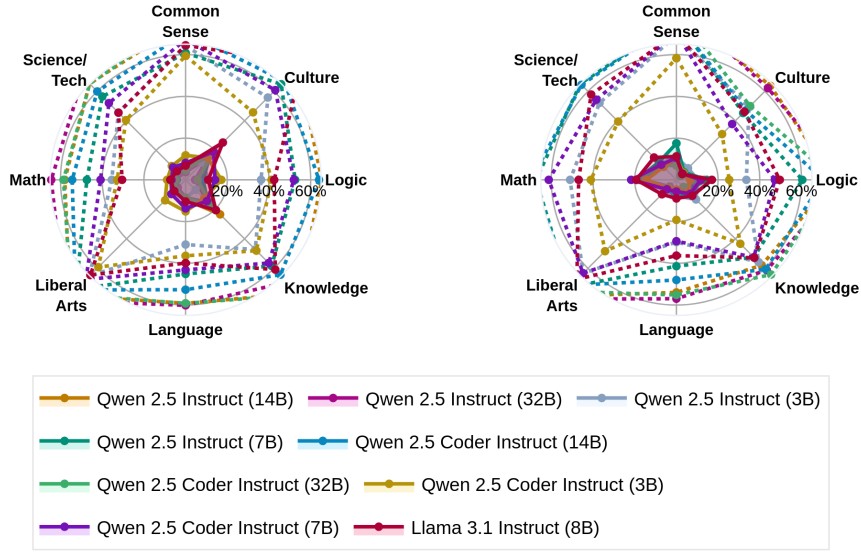

Figure 21: **Forgetting (left) and Backward Transfer (right) after instruction-tuning.** Dotted (solid) lines represent maximum (empirical) values.

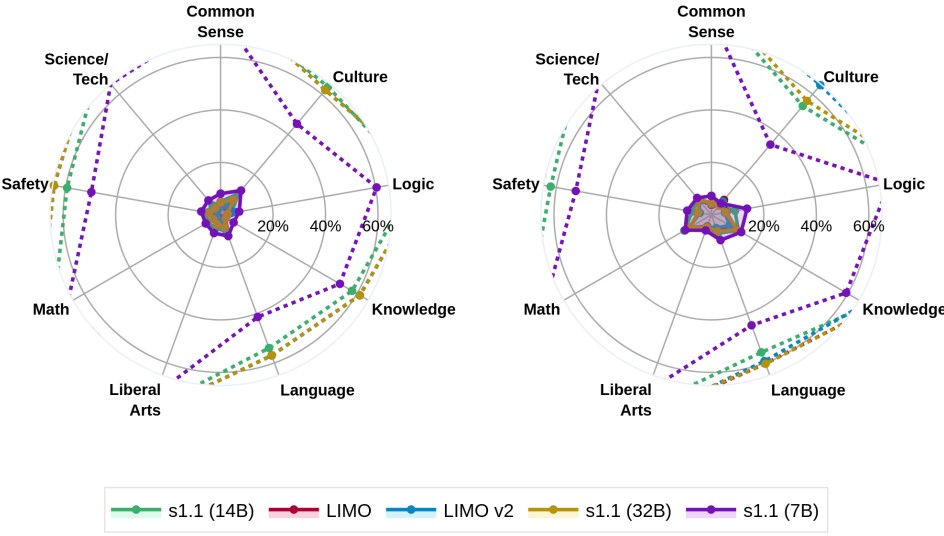

Figure 22: **Forgetting (left) and Backward Transfer (right) after reasoning training from instruct: low data scenario.** Dotted (solid) lines represent maximum (empirical) values.

## F    MODEL MERGING ANALYSIS

This section provides full radar charts and qualitative analysis for the model merging experiments. We evaluate linear interpolation between post-trained and base (or instruct) checkpoints, examining three cases: two outright failures (OpenThinker3 and Qwen 2.5 Coder) and one moderate success (OpenThinker). The weight-drift analysis in §F.1 offers a unifying explanation for why merging succeeds or fails.

### F.1    WEIGHT DRIFT

We observe large weight drift among models in which merging fails. Specifically in the case of model trained from instruction tuned bases, we compute the ratio of the $L_2$ norm of the task vector (from the model to the instruct model) to the $L_2$ norm of the base model. In the case of OpenThinker3 this is just above 20%. Likewise Qwen2.5 Coder (7B) has a value of 87%. OpenThinker and s1.1, which we find are both mergeable, have values of only 1.8% and 0.6%, respectively.

### F.2 FAILURE CASE: OPENTHINKER3

OpenThinker3 is a 7B reasoning model derived from Qwen 2.5 Instruct via large-scale RL post-training. Despite sharing the same base, its task vector has an $L_2$ norm ratio of approximately $20\%$ relative to the base model, roughly ten times that of the mergeable OpenThinker (F.1). Figures 23 and 24 evaluate the merged checkpoint relative to Qwen 2.5 Instruct and to OpenThinker3 itself, respectively. Both evaluations reveal large forgetting across all categories. Sample-level inspection shows the merged model frequently degenerates into repetitive output, looping over words, phrases, or numerical sequences without producing a coherent final answer. This behavioral collapse, in contrast to a gradual redistribution of capabilities, suggests that the large task vector overwhelms the interpolation, displacing the checkpoint into a region of the weight space that supports neither model's behavior.

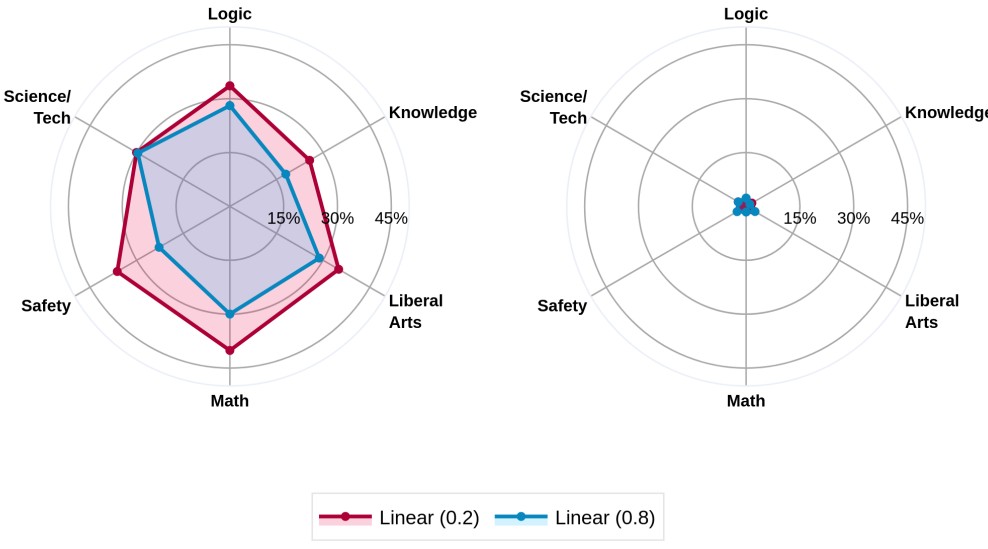

Figure 23: **Forgetting (left) and Backward Transfer (right) of Qwen 2.5 Instruct merged with OpenThinker3 (7B) relative to Qwen 2.5 Instruct on MMLU**. Large forgetting occurs. Sample-level analysis shows the model output degeneration, with the model often repeating words or phrases, typically without providing a final answer.

Figure 23 shows that the degradation is not confined to a single knowledge cluster, but occurs broadly, spanning commonsense, culture, language, and liberal arts knowledge categories. This broad degradation is distinct from ordinary domain forgetting, in which one would expect forgetting to concentrate in areas semantically distant from the training domain. Instead, the merged checkpoint's failure is structural. Selective forgetting suggests a poor weight selection, whereas output collapse suggests the merged weights have landed outside the basin of any well-defined behavior. Figure 24 confirms the same pattern when evaluated from the perspective of OpenThinker3 to the merged model.

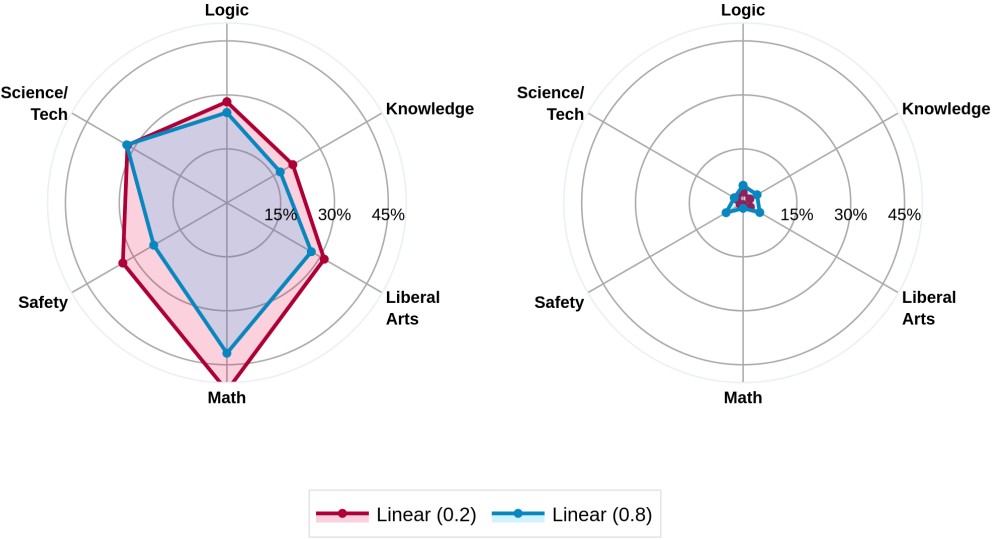

Figure 24: **Forgetting (left) and Backward Transfer (right) of Qwen 2.5 Instruct merged with OpenThinker3 (7B) relative to OpenThinker3 on MMLU**. Large forgetting occurs. Sample-level analysis shows the model output degeneration, with the model often repeating words or phrases, typically without providing a final answer.

### F.3 FAILURE CASE: CODER MODELS

Qwen 2.5 Coder (7B) presents the most extreme weight drift in our study. Its task vector has an $L_2$ norm ratio of $87\%$ relative to the base model, nearly equal in magnitude to the base weights themselves. This reflects the depth of domain-continual pretraining. The model has been substantially reshaped from Qwen 2.5 Base to specialize in code generation. Merging it back with the base therefore requires interpolating between two very dissimilar weight configurations.

Figures 25 and 26 show the result relative to Qwen 2.5 Base and to Qwen 2.5 Coder, respectively. Both comparisons show moderate-to-large forgetting with low backward transfer throughout. The merge fails to consolidate the two checkpoints; the resulting model neither retains the base model's broad knowledge nor preserves the coder's specialized representations. Instead it occupies a degraded intermediate state, illustrating that extreme weight drift makes interpolation destructive rather than complementary.

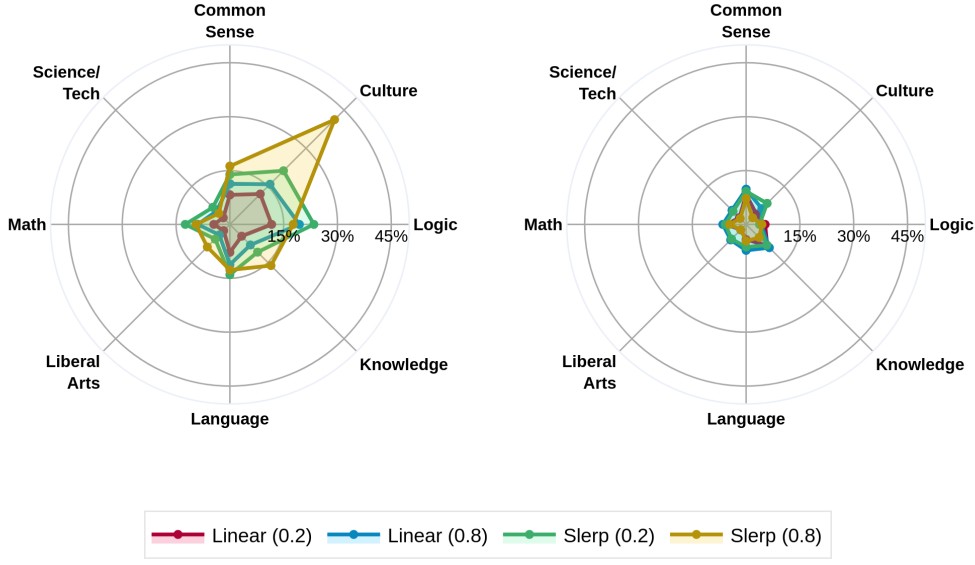

Figure 25: **Forgetting (left) and Backward Transfer (right) of Qwen 2.5 Base merged with Qwen 2.5 Coder (7B) relative to Qwen 2.5 Base on all Benchmarks**. Moderate-to-large forgetting occurs with low backward transfer.

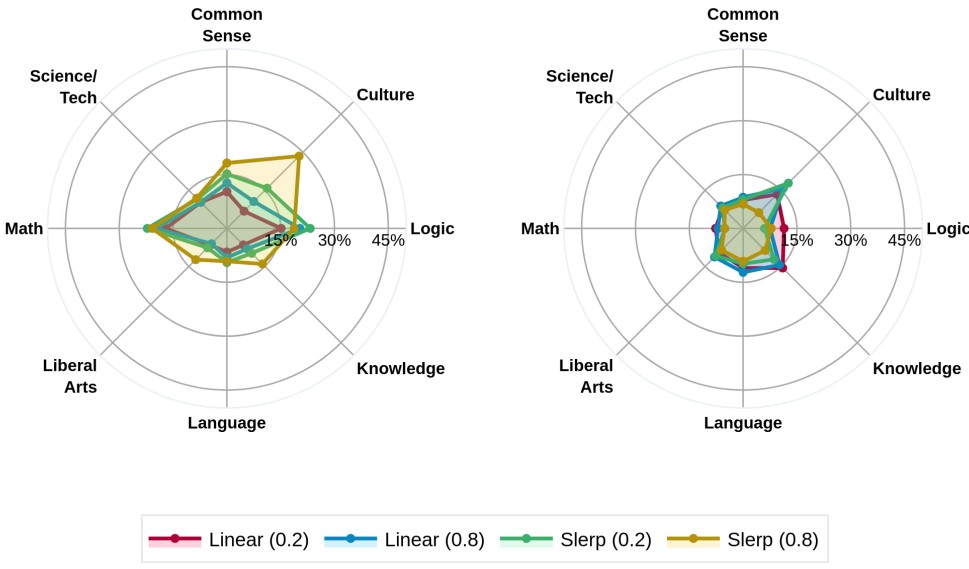

Figure 26: **Forgetting (left) and Backward Transfer (right) of Qwen 2.5 Base merged with Qwen 2.5 Coder (7B) relative to Qwen 2.5 Coder on all Benchmarks**. Moderate-to-large forgetting occurs with low-to-moderate backward transfer.

### F.4 MODERATE CASE: OPENTHINKER

OpenThinker (7B) provides a contrasting case to the two failure cases above. Its task vector $L_2$ norm ratio is only $1.8\%$, indicating that SFT on reasoning traces made comparatively minor weight adjustments relative to Qwen 2.5 Instruct. With much smaller weight drift, the two checkpoints are more geometrically compatible, and interpolation is less disruptive.

Figures 27 and 28 show the results relative to Qwen 2.5 Instruct and to OpenThinker, respectively. Both evaluations show only modest forgetting, and the Linear (0.8) interpolation yields a marginal

overall improvement relative to either endpoint. This is consistent with the weight-drift hypothesis, that models small changes from their initialization are more amenable to interpolation.

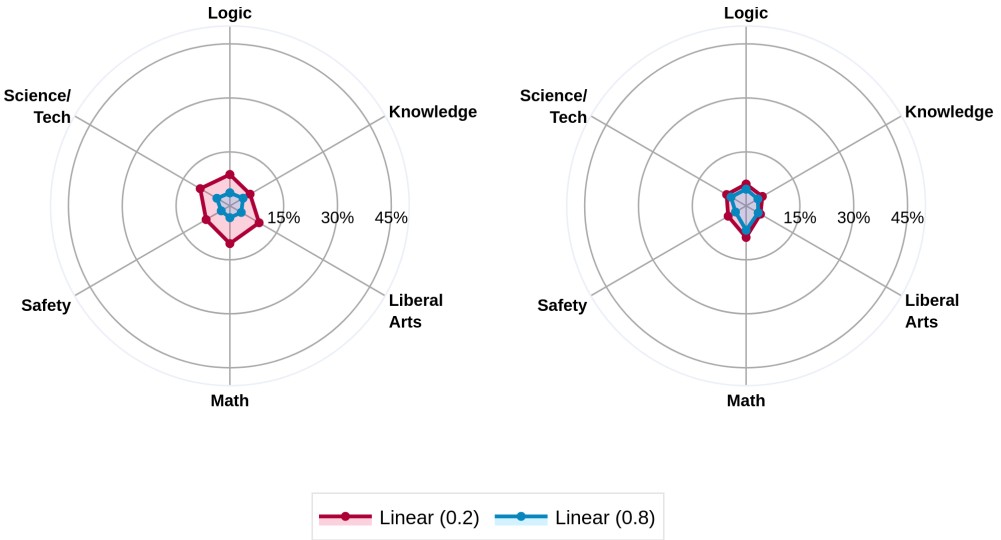

Figure 27: **Forgetting (left) and Backward Transfer (right) of Qwen 2.5 Instruct merged with OpenThinker Merge (7B) relative to Qwen 2.5 Instruct on MMLU**. We see a marginal overall performance improvement in the case of Linear (0.8).

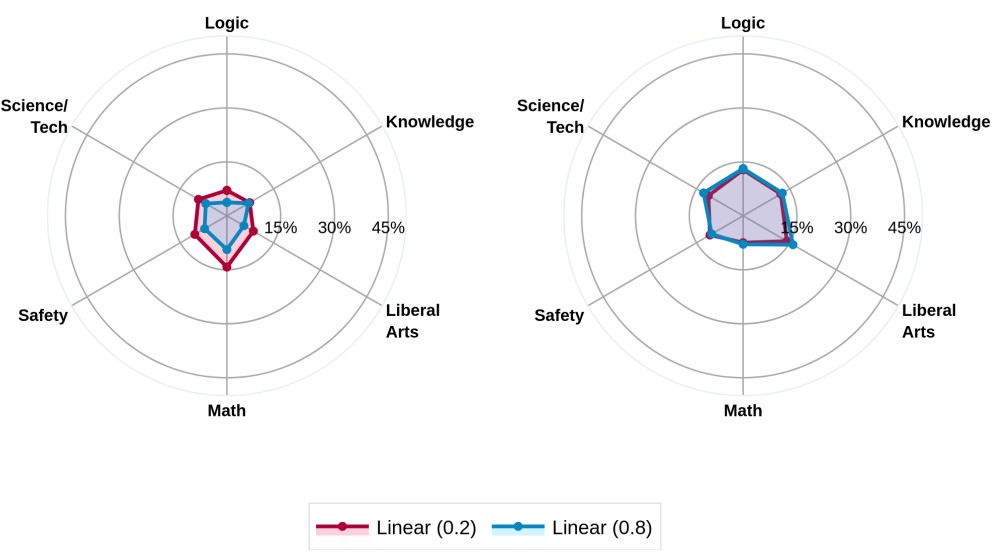

Figure 28: **Forgetting (left) and Backward Transfer (right) of Qwen 2.5 Instruct merged with OpenThinker (7B) relative to OpenThinker on MMLU**. We see a marginal overall performance improvement in both cases.

Figures 27 and 28 reveal that even in the moderate-success case, the improvement is concentrated in the Linear (0.8) interpolation and is small in absolute terms. Heavier interpolation toward the base checkpoint progressively erodes reasoning gains without recovering broad knowledge proportionally.

# G PROMPTS

This section provides the exact prompt templates used throughout the evaluation pipeline. The instruction-tuned template (§G.1) is used for models that have undergone instruction tuning; the few-shot base template (§G.2) is used for base models that require in-context demonstrations; and the LLM extractor template (§G.3) is used to parse free-form model outputs to account for regex-based extraction fails.

## G.1 INSTRUCTION-TUNED PROMPT TEMPLATE

```
{Instruction}

On the very last line, write exactly "Answer: $LETTER" (e.g.
"Answer: B"), with no extra punctuation, no lowercase, no *,
and no trailing spaces.
Think step by step, showing your reasoning.
Question: "{Question}"
```

## G.2 BASE MODEL PROMPT TEMPLATE

```
{Instruction}

Question: "{Few-Shot Question 1}"
Reasoning: {Few-shot Reasoning Trace 1}
Answer: {Few-shot Answer 1}

... <--- more examples

Question: "{Question}"
Reasoning:
```

## G.3 LLM EXTRACTOR TEMPLATE

```
You are a strict extractor.
Given the FULL_PROMPT (the original prompt to the model), the model
output SNIPPET (last part, quoted), and the gold extraction (quoted),
return ONLY valid JSON with exactly two keys:
  - "extraction": the final answer token as a string
    (e.g. "A", "C", "42") or null if unknown
  - "correct": true if the extraction matches  the gold, false
    if it does not, or null if unknown
Do NOT output anything else (no explanation, no code fences).

FULL_PROMPT: {q_full_prompt}
SNIPPET: {q_snippet}
GOLD: {q_gold}

Return JSON now:
```

## H  DATASET GROUPINGS

The following taxonomy groups the evaluated benchmarks into nine thematic categories. These groupings serve as the backbone for the forgetting and backward transfer radar charts throughout the paper [5].

**Commonsense**

- Commonsense QA
- PIQA

**Culture**

- BBH (sports understanding, movie recommendation)

**Logic**

- BBH (navigate, causal judgment, penguins in a table, web of lies, tracking shuffled objects three objects, tracking shuffled objects seven objects, tracking shuffled objects five objects, temporal sequences, reasoning about colored objects, logical deduction three objects, logical deduction seven objects, logical deduction five objects, formal fallacies, and date understanding)
- ARC (easy and challenge)
- MuSR (murder mysteries, object placements, team allocation)
- MMLU (logical fallacies)

**Knowledge**

- BBH (object counting)
- MMLU (miscellaneous, global facts)
- MCTest

**Language**

- BBH (snarks, disambiguation qa, ruin names, hyperbaton, translation error detection)
- Social IQa, Hellaswag

**Math**

- BBH (geometric shapes, and boolean expressions)
- MMLU (high school statistics, high school mathematics, formal logic, elementary mathematics, econometrics, college mathematics, and abstract algebra)

**Math (Generative)**

- AIME24, AIME25, AMC23
- Math500, Minerva, OlympiadBench

**Safety** [6]

- MMLU (moral scenarios, moral disputes, jurisprudence, and business ethics)
- TruthfulQA (mc1)
- SaladBench (mrq)

**Science & Tech**

- MMLU (marketing, virology, professional medicine, professional accounting, nutrition, medical genetics, machine learning, human sexuality, human aging, high school physics, high school computer science, high school chemistry, high school biology, electrical engineering, conceptual physics, computer security, college physics, college medicine, college computer science, college chemistry, college biology, clinical knowledge, astronomy, and anatomy)
- GPQA (diamond)

**Liberal Arts**

- MMLU (world religions, us foreign policy, sociology, security studies, public relations, professional psychology, professional law, prehistory, philosophy, management, international law, high school world history, high school us history, high school psychology, high school microeconomics, high school macroeconomics, high school government and politics, high school geography, and high school european history)

---

[5]MMLU is evaluated with few-shot, no CoT prompting for the base models.

[6]These are only used in comparisons that do not include a base model because TruthfulQA and SaladBench are designed to measure the default behavior of the model rather than knowledge, which few-shot prompting would bias.

# I  TABLES

The following tables provide numerical counterparts to the radar charts in the main paper and in §D, covering all post-training conditions and model families evaluated. Each cell reports chance-adjusted forgetting (or backward transfer) as a percentage, the standard deviation across seeds after "$\pm$", and the maximum possible value in parentheses. For example, $12.3_{\pm 0.4} (67.8)$ denotes $12.3\%$ forgetting with standard deviation $0.4\%$ and a ceiling of $67.8\%$. Forgetting and backward transfer tables are paired within each subsection. The first five subsections cover the post-training regimes analyzed in §3; the remaining ones provide numerical detail for the merging experiments discussed in §3 and §F.1.

## I.1  INSTRUCTION TUNING

Numerical results corresponding to Figure 4, covering Qwen 2.5 Instruct (3B–32B), Qwen 2.5 Coder Instruct (3B–32B), and Llama 3.1 Instruct (8B).

Table 2: Instruction Tuning: Forgetting (Part 1 of 3)

| Category | Qwen 2.5 Instruct | | | |
| --- | --- | --- | --- | --- |
| | 3B | 7B | 14B | 32B |
| Common Sense | $7.0_{\pm 0.3} (64.2)$ | $5.4_{\pm 0.1} (60.8)$ | $3.9_{\pm 0.6} (75.8)$ | $3.7_{\pm 0.5} (79.5)$ |
| Culture | $11.7_{\pm 0.9} (55.8)$ | $16.5_{\pm 2.8} (64.8)$ | $14.0_{\pm 0.1} (76.4)$ | $15.2_{\pm 0.6} (78.9)$ |
| Logic | $10.9_{\pm 0.5} (36.2)$ | $8.4_{\pm 0.2} (52.5)$ | $5.5_{\pm 0.4} (65.3)$ | $4.6_{\pm 0.6} (74.4)$ |
| Knowledge/QA | $6.8_{\pm 1.3} (47.3)$ | $15.0_{\pm 1.0} (58.4)$ | $23.4_{\pm 0.3} (76.9)$ | $15.3_{\pm 1.9} (69.4)$ |
| Language | $8.7_{\pm 0.8} (31.0)$ | $9.2_{\pm 0.6} (45.1)$ | $9.5_{\pm 0.8} (59.2)$ | $8.6_{\pm 1.0} (60.2)$ |
| Liberal Arts | $8.7_{\pm 0.7} (65.3)$ | $6.6_{\pm 0.7} (74.2)$ | $5.3_{\pm 0.4} (78.6)$ | $5.3_{\pm 0.3} (81.9)$ |
| Math | $7.7_{\pm 0.4} (35.4)$ | $4.2_{\pm 0.4} (47.3)$ | $6.2_{\pm 0.7} (57.9)$ | $4.5_{\pm 1.5} (64.4)$ |
| Math (Gen) | $2.5_{\pm 0.4} (8.4)$ | $2.6_{\pm 1.0} (12.8)$ | $2.8_{\pm 0.6} (14.5)$ | $3.9_{\pm 1.7} (18.7)$ |
| Science/Tech | $6.5_{\pm 0.3} (45.6)$ | $5.4_{\pm 0.4} (56.5)$ | $4.5_{\pm 0.5} (65.2)$ | $4.4_{\pm 0.3} (69.7)$ |
| **Total** | $8.5_{\pm 0.1} (50.2)$ | $8.2_{\pm 0.3} (58.6)$ | $8.0_{\pm 0.3} (69.7)$ | $6.7_{\pm 0.5} (72.3)$ |

Table 3: Instruction Tuning: Forgetting (Part 2 of 3)

| Category | Qwen 2.5 Coder Instruct | | | |
| --- | --- | --- | --- | --- |
| | 3B | 7B | 14B | 32B |
| Common Sense | $11.7_{\pm 1.0} (59.5)$ | $8.1_{\pm 0.3} (67.2)$ | $6.1_{\pm 0.2} (70.7)$ | $4.6_{\pm 0.4} (77.9)$ |
| Culture | $15.0_{\pm 2.6} (45.7)$ | $19.0_{\pm 1.9} (60.8)$ | $16.6_{\pm 1.6} (66.9)$ | $19.1_{\pm 1.1} (73.8)$ |
| Logic | $17.2_{\pm 0.6} (41.2)$ | $14.1_{\pm 0.2} (51.9)$ | $6.9_{\pm 0.2} (64.0)$ | $5.8_{\pm 0.3} (69.8)$ |
| Knowledge/QA | $12.6_{\pm 0.1} (48.0)$ | $14.4_{\pm 0.3} (56.4)$ | $14.3_{\pm 0.3} (64.3)$ | $17.6_{\pm 1.0} (77.7)$ |
| Language | $15.1_{\pm 0.7} (36.4)$ | $13.7_{\pm 1.0} (43.2)$ | $10.3_{\pm 0.8} (52.7)$ | $8.6_{\pm 0.5} (59.2)$ |
| Liberal Arts | $13.9_{\pm 0.6} (59.2)$ | $9.6_{\pm 0.0} (67.6)$ | $7.8_{\pm 0.4} (74.6)$ | $6.7_{\pm 0.2} (77.7)$ |
| Math | $8.9_{\pm 0.9} (32.8)$ | $6.8_{\pm 0.2} (40.8)$ | $6.2_{\pm 0.4} (54.4)$ | $4.9_{\pm 0.6} (58.6)$ |
| Math (Gen) | $2.8_{\pm 0.3} (7.8)$ | $3.6_{\pm 1.1} (10.6)$ | $3.0_{\pm 0.7} (11.8)$ | $2.8_{\pm 0.4} (13.4)$ |
| Science/Tech | $9.4_{\pm 0.7} (40.5)$ | $8.4_{\pm 0.5} (52.0)$ | $7.0_{\pm 0.2} (59.9)$ | $5.6_{\pm 0.5} (65.1)$ |
| **Total** | $13.0_{\pm 0.2} (48.2)$ | $10.9_{\pm 0.3} (57.6)$ | $8.9_{\pm 0.3} (66.2)$ | $8.4_{\pm 0.1} (72.2)$ |

Table 4: Instruction Tuning: Forgetting (Part 3 of 3)

| Category | Llama 3.1 Instruct (8B) |
|---|---|
| Common Sense | 6.9 ±0.4 (64.5) |
| Culture | 25.3 ±2.2 (79.1) |
| Logic | 10.9 ±0.5 (42.5) |
| Knowledge/QA | 20.6 ±0.8 (60.8) |
| Language | 10.4 ±0.7 (39.9) |
| Liberal Arts | 7.6 ±0.6 (64.6) |
| Math | 7.3 ±0.9 (30.5) |
| Math (Gen) | 1.6 ±0.3 (3.7) |
| Science/Tech | 5.9 ±0.9 (45.5) |
| **Total** | 10.8 ±0.3 (54.5) |

Table 5: Instruction Tuning: Backward Transfer (Part 1 of 3)

| Category | Qwen 2.5 Instruct | | | |
|---|---|---|---|---|
| | 3B | 7B | 14B | 32B |
| Common Sense | 10.8 ±0.4 (69.2) | 17.5 ±0.3 (76.9) | 8.7 ±0.1 (82.1) | 7.5 ±0.3 (84.6) |
| Culture | 7.9 ±2.4 (49.2) | 5.2 ±1.9 (45.3) | 5.3 ±0.3 (63.7) | 3.0 ±1.1 (62.1) |
| Logic | 10.2 ±0.7 (33.6) | 14.6 ±0.5 (60.4) | 13.3 ±0.5 (75.6) | 9.0 ±0.3 (80.4) |
| Knowledge/QA | 13.3 ±2.8 (55.8) | 7.5 ±2.0 (52.5) | 5.1 ±1.0 (57.9) | 7.7 ±1.7 (61.3) |
| Language | 7.7 ±0.5 (29.8) | 8.3 ±0.4 (41.4) | 7.2 ±0.0 (54.0) | 8.3 ±2.4 (57.0) |
| Liberal Arts | 7.2 ±1.4 (63.3) | 5.7 ±0.6 (73.0) | 5.6 ±0.9 (78.9) | 4.6 ±1.1 (80.9) |
| Math | 18.9 ±0.9 (51.0) | 19.0 ±0.8 (67.3) | 17.8 ±1.4 (73.8) | 15.9 ±1.9 (80.0) |
| Math (Gen) | 14.9 ±0.9 (20.8) | 18.7 ±1.4 (28.9) | 21.6 ±1.0 (33.3) | 18.6 ±1.9 (33.4) |
| Science/Tech | 11.4 ±0.8 (52.1) | 11.9 ±0.9 (65.2) | 10.4 ±1.0 (73.0) | 9.5 ±0.8 (76.5) |
| **Total** | 10.5 ±0.5 (52.7) | 11.0 ±0.5 (62.1) | 8.9 ±0.4 (71.3) | 8.2 ±0.7 (74.3) |

Table 6: Instruction Tuning: Backward Transfer (Part 2 of 3)

| Category | Qwen 2.5 Coder Instruct | | | |
|---|---|---|---|---|
| | 3B | 7B | 14B | 32B |
| Common Sense | 10.9 ±0.1 (58.3) | 10.6 ±1.0 (70.5) | 10.2 ±0.8 (76.2) | 6.9 ±0.1 (80.9) |
| Culture | 4.8 ±0.9 (31.0) | 4.2 ±0.7 (37.8) | 2.8 ±0.5 (47.0) | 2.7 ±0.7 (50.0) |
| Logic | 6.6 ±0.3 (25.3) | 11.7 ±0.7 (47.2) | 11.5 ±0.3 (70.6) | 10.8 ±0.1 (76.4) |
| Knowledge/QA | 8.3 ±1.2 (43.4) | 9.4 ±1.9 (52.5) | 10.2 ±1.3 (60.6) | 4.8 ±1.5 (64.2) |
| Language | 4.7 ±0.4 (19.2) | 5.6 ±1.2 (29.4) | 7.9 ±0.2 (48.0) | 6.5 ±0.7 (55.2) |
| Liberal Arts | 5.8 ±1.1 (48.4) | 6.4 ±0.7 (63.3) | 5.5 ±1.1 (71.6) | 5.1 ±0.8 (75.6) |
| Math | 14.4 ±1.8 (41.0) | 21.7 ±1.4 (61.2) | 17.5 ±1.9 (69.6) | 19.1 ±1.5 (77.8) |
| Math (Gen) | 9.3 ±1.3 (14.3) | 14.7 ±2.6 (21.7) | 20.3 ±2.6 (29.1) | 24.1 ±1.3 (34.7) |
| Science/Tech | 8.7 ±1.6 (39.5) | 10.2 ±0.5 (54.3) | 10.4 ±1.0 (64.4) | 9.8 ±0.4 (70.6) |
| **Total** | 7.4 ±0.2 (40.4) | 9.3 ±0.6 (54.9) | 8.6 ±0.3 (65.8) | 7.6 ±0.4 (71.2) |

Table 7: Instruction Tuning: Backward Transfer (Part 3 of 3)

| Category | Llama 3.1 Instruct (8B) |
|---|---|
| Common Sense | 11.2 ±0.3 (70.3) |
| Culture | 3.9 ±0.2 (46.2) |
| Logic | 17.0 ±0.9 (49.6) |
| Knowledge/QA | 10.8 ±2.3 (52.9) |
| Language | 8.9 ±1.4 (36.3) |
| Liberal Arts | 9.8 ±1.7 (67.5) |
| Math | 19.3 ±0.7 (46.9) |
| Math (Gen) | 11.8 ±0.5 (14.0) |
| Science/Tech | 15.1 ±1.3 (57.8) |
| **Total** | 11.4 ±1.0 (55.1) |

## I.2 DOMAIN-CONTINUAL PRETRAINING

Numerical results corresponding to Figure 3, covering Qwen 2.5 Coder (3B–32B) and Qwen 2.5 Math (7B).

Table 8: Domain-Continual Pretraining: Forgetting (Part 1 of 2)

| Category | Qwen 2.5 Coder | | | |
|---|---|---|---|---|
| | 3B | 7B | 14B | 32B |
| Common Sense | 11.9 ±0.5 (64.2) | 9.0 ±0.4 (60.8) | 10.4 ±0.6 (75.8) | 7.5 ±0.6 (79.5) |
| Culture | 11.7 ±0.7 (55.8) | 10.9 ±0.4 (64.8) | 10.8 ±0.7 (76.4) | 8.8 ±0.8 (78.9) |
| Logic | 5.7 ±0.2 (36.2) | 9.5 ±0.2 (52.5) | 7.6 ±0.3 (65.3) | 8.4 ±0.1 (74.4) |
| Knowledge/QA | 5.6 ±0.5 (47.3) | 6.0 ±0.8 (58.4) | 13.7 ±0.2 (76.9) | 3.8 ±0.5 (69.4) |
| Language | 5.9 ±0.6 (31.0) | 8.4 ±1.5 (45.1) | 8.6 ±0.9 (59.2) | 7.4 ±1.3 (60.2) |
| Liberal Arts | 6.4 ±0.5 (65.3) | 7.0 ±0.3 (74.2) | 5.1 ±0.4 (78.6) | 5.2 ±0.3 (81.9) |
| Math | 3.8 ±0.9 (35.4) | 7.4 ±1.0 (47.3) | 6.2 ±0.2 (57.9) | 7.8 ±0.5 (64.4) |
| Math (Gen) | 4.6 ±0.2 (8.4) | 7.1 ±1.1 (12.8) | 8.3 ±1.0 (14.5) | 10.0 ±2.3 (18.7) |
| Science/Tech | 4.0 ±0.5 (45.6) | 5.9 ±0.3 (56.5) | 6.4 ±0.5 (65.2) | 5.7 ±0.5 (69.7) |
| **Total** | 6.8 ±0.0 (50.9) | 7.6 ±0.2 (60.4) | 8.0 ±0.1 (71.8) | 6.4 ±0.2 (74.6) |

Table 9: Domain-Continual Pretraining: Forgetting (Part 2 of 2)

| Category | Qwen 2.5 Math (7B) |
|---|---|
| Common Sense | 13.6 ±0.8 (60.8) |
| Culture | 17.8 ±0.3 (64.8) |
| Logic | 9.8 ±0.4 (52.5) |
| Knowledge/QA | 9.8 ±0.5 (58.4) |
| Language | 11.3 ±0.9 (45.1) |
| Liberal Arts | 20.0 ±1.3 (74.2) |
| Math | 7.5 ±1.2 (47.3) |
| Math (Gen) | 7.0 ±1.3 (12.8) |
| Science/Tech | 14.4 ±0.5 (56.5) |
| **Total** | 12.9 ±0.4 (60.4) |

Table 10: Domain-Continual Pretraining: Backward Transfer (Part 1 of 2)

| | Qwen 2.5 Coder | | | |
|---|---|---|---|---|
| Category | 3B | 7B | 14B | 32B |
| Common Sense | 8.3 ±0.4 (59.5) | 13.8 ±0.3 (67.2) | 6.6 ±0.3 (70.7) | 6.3 ±0.5 (77.9) |
| Culture | 6.3 ±2.1 (45.7) | 10.0 ±1.3 (60.8) | 5.8 ±0.4 (66.9) | 5.6 ±0.2 (73.8) |
| Logic | 9.4 ±0.5 (41.2) | 9.7 ±0.2 (51.9) | 7.7 ±0.8 (64.0) | 5.3 ±0.4 (69.8) |
| Knowledge/QA | 6.6 ±1.1 (48.0) | 5.4 ±0.5 (56.4) | 2.7 ±0.3 (64.3) | 12.4 ±0.7 (77.7) |
| Language | 8.9 ±0.9 (36.4) | 7.6 ±0.8 (43.2) | 5.3 ±0.5 (52.7) | 7.8 ±2.2 (59.2) |
| Liberal Arts | 2.0 ±0.2 (59.2) | 2.2 ±0.3 (67.6) | 2.3 ±0.3 (74.6) | 2.1 ±0.1 (77.7) |
| Math | 2.4 ±0.2 (32.8) | 3.2 ±0.3 (40.8) | 3.4 ±1.2 (54.4) | 3.4 ±1.0 (58.6) |
| Math (Gen) | 4.0 ±0.9 (7.8) | 4.9 ±0.9 (10.6) | 5.7 ±0.9 (11.8) | 4.7 ±0.1 (13.4) |
| Science/Tech | 1.0 ±0.3 (40.5) | 2.8 ±0.0 (52.0) | 2.6 ±0.1 (59.9) | 2.4 ±0.2 (65.1) |
| **Total** | 5.1 ±0.2 (48.2) | 6.2 ±0.1 (57.6) | 4.2 ±0.2 (66.2) | 5.1 ±0.2 (72.2) |

Table 11: Domain-Continual Pretraining: Backward Transfer (Part 2 of 2)

| Category | Qwen 2.5 Math (7B) |
|---|---|
| Common Sense | 12.7 ±0.5 (59.6) |
| Culture | 6.9 ±0.8 (46.3) |
| Logic | 11.9 ±0.2 (53.9) |
| Knowledge/QA | 9.9 ±1.8 (55.3) |
| Language | 6.1 ±0.2 (36.3) |
| Liberal Arts | 1.6 ±0.3 (49.5) |
| Math | 4.3 ±0.5 (43.1) |
| Math (Gen) | 4.5 ±0.8 (10.3) |
| Science/Tech | 3.2 ±0.7 (40.7) |
| **Total** | 6.3 ±0.2 (50.1) |

## I.3 TRAINED FROM BASE

Numerical results corresponding to Figure 5, covering SFT/RL reasoning training from base model checkpoints: Qwen 2.5 Math Instruct (7B), QwQ-32B, DeepSeek-R1-Distill Qwen (7B and 32B), and DeepSeek-R1-Distill Llama (8B).

Table 12: Trained from Base: Forgetting (Part 1 of 2)

| | Qwen | | R1 Distill Qwen | |
|---|---|---|---|---|
| Category | v2.5 Math Inst. (7B) | QwQ (32B) | 7B | 32B |
| Common Sense | 26.0 ±0.3 (59.6) | 3.2 ±0.4 (79.5) | 8.9 ±0.3 (59.6) | 3.9 ±0.3 (79.5) |
| Culture | 27.0 ±2.4 (46.3) | 15.7 ±0.6 (78.9) | 19.3 ±2.1 (46.3) | 18.8 ±0.7 (78.9) |
| Logic | 16.8 ±0.4 (53.9) | 2.8 ±0.2 (74.4) | 4.4 ±0.4 (53.9) | 2.3 ±0.3 (74.4) |
| Knowledge/QA | 25.4 ±2.0 (55.3) | 13.2 ±2.1 (69.4) | 20.1 ±0.7 (55.3) | 15.3 ±1.3 (69.4) |
| Language | 18.2 ±0.9 (36.3) | 7.9 ±1.2 (60.2) | 9.4 ±0.8 (36.3) | 6.9 ±1.4 (60.2) |
| Liberal Arts | 23.7 ±0.7 (49.5) | 3.1 ±0.2 (81.9) | 7.6 ±0.4 (49.5) | 3.6 ±0.3 (81.9) |
| Math | 13.1 ±0.3 (43.1) | 1.5 ±0.3 (64.4) | 2.2 ±0.4 (43.1) | 1.8 ±0.5 (64.4) |
| Math (Gen) | 2.1 ±0.4 (10.3) | 1.1 ±0.2 (18.7) | 0.8 ±0.3 (10.3) | 1.3 ±0.2 (18.7) |
| Science/Tech | 17.1 ±0.9 (40.7) | 2.2 ±0.2 (69.7) | 4.6 ±0.2 (40.7) | 2.4 ±0.3 (69.7) |
| **Total** | 21.4 ±0.5 (50.1) | 5.4 ±0.3 (72.3) | 8.7 ±0.1 (50.1) | 6.0 ±0.2 (72.3) |

Table 13: Trained from Base: Forgetting (Part 2 of 2)

| Category | R1 Distill Llama (8B) |
|---|---|
| Common Sense | 8.1 ±0.2 (64.5) |
| Culture | 29.0 ±1.0 (79.1) |
| Logic | 4.8 ±0.2 (42.5) |
| Knowledge/QA | 11.7 ±0.7 (60.8) |
| Language | 10.4 ±0.4 (39.9) |
| Liberal Arts | 9.4 ±0.9 (64.6) |
| Math | 3.3 ±0.3 (30.5) |
| Math (Gen) | 0.6 ±0.2 (3.7) |
| Science/Tech | 6.4 ±0.4 (45.5) |
| **Total** | 9.3 ±0.3 (54.5) |

Table 14: Trained from Base: Backward Transfer (Part 1 of 2)

| Category | **Qwen** | | **R1 Distill Qwen** | |
|---|---|---|---|---|
| | v2.5 Math Inst. (7B) | QwQ (32B) | 7B | 32B |
| Common Sense | 6.0 ±0.6 (32.9) | 8.9 ±0.5 (87.0) | 11.4 ±0.1 (63.0) | 8.7 ±0.3 (85.9) |
| Culture | 3.4 ±1.3 (9.0) | 2.9 ±1.0 (59.8) | 5.6 ±1.4 (23.1) | 2.5 ±0.6 (55.4) |
| Logic | 8.4 ±0.4 (42.8) | 11.8 ±0.5 (86.7) | 19.7 ±0.1 (74.9) | 11.7 ±0.4 (87.2) |
| Knowledge/QA | 4.1 ±1.4 (31.2) | 9.0 ±1.6 (65.2) | 9.3 ±2.4 (45.6) | 6.8 ±1.3 (60.6) |
| Language | 9.6 ±0.6 (20.9) | 9.7 ±2.8 (33.7) | 8.1 ±0.8 (33.7) | 10.2 ±2.7 (62.9) |
| Liberal Arts | 5.0 ±0.4 (24.5) | 6.3 ±1.0 (86.1) | 12.7 ±1.6 (56.2) | 6.1 ±1.0 (85.2) |
| Math | 15.1 ±1.6 (46.5) | 22.5 ±2.2 (92.2) | 34.3 ±2.4 (85.8) | 22.1 ±2.5 (91.4) |
| Math (Gen) | 23.6 ±3.1 (31.7) | 54.8 ±1.3 (72.3) | 50.5 ±2.2 (60.0) | 50.3 ±2.3 (67.7) |
| Science/Tech | 7.2 ±0.2 (27.4) | 13.8 ±1.0 (85.0) | 18.5 ±1.5 (59.3) | 13.3 ±1.0 (84.2) |
| **Total** | 6.7 ±0.2 (30.1) | 10.3 ±0.6 (78.6) | 14.3 ±1.0 (57.4) | 10.0 ±0.6 (77.5) |

Table 15: Trained from Base: Backward Transfer (Part 2 of 2)

| Category | R1 Distill Llama (8B) |
|---|---|
| Common Sense | 12.6 ±0.8 (70.5) |
| Culture | 2.8 ±0.4 (36.5) |
| Logic | 28.9 ±0.1 (74.6) |
| Knowledge/QA | 10.8 ±1.8 (61.8) |
| Language | 12.1 ±1.0 (42.2) |
| Liberal Arts | 10.3 ±1.5 (65.8) |
| Math | 35.0 ±0.5 (72.8) |
| Math (Gen) | 49.2 ±1.5 (52.4) |
| Science/Tech | 19.3 ±1.2 (62.7) |
| **Total** | 15.4 ±0.6 (62.1) |

## I.4  TRAINED FROM INSTRUCT - HIGH DATA SCENARIO

Numerical results corresponding to Figure 6, covering large-scale SFT/RL reasoning post-training from instruction-tuned models: OpenThinker (7B), OpenThinker2 (32B), OpenThinker3 (7B), Open-CodeReasoner, OpenMath2, and Intellect-2 (32B).

Table 16: Trained from Instruct - High Data Scenario: Forgetting (Part 1 of 2)

| Category | INTELLECT-2 | OpenThinker | | |
| | 32B | v1 (7B) | v2 (32B) | v3 (7B) |
|---|---|---|---|---|
| Common Sense | 1.9 ±0.3 (87.0) | 18.1 ±1.3 (76.9) | 3.9 ±0.3 (84.6) | 9.5 ±0.4 (76.9) |
| Culture | 0.7 ±0.6 (60.2) | 13.3 ±4.6 (45.3) | 7.6 ±2.1 (62.1) | 16.4 ±4.5 (45.3) |
| Logic | 0.6 ±0.1 (87.4) | 7.4 ±0.2 (60.4) | 1.0 ±0.2 (80.4) | 5.2 ±0.2 (60.4) |
| Knowledge/QA | 2.5 ±0.6 (65.9) | 13.4 ±1.1 (52.5) | 2.8 ±0.5 (61.3) | 5.6 ±0.9 (52.5) |
| Language | 1.9 ±0.3 (62.7) | 10.7 ±1.2 (41.4) | 4.2 ±0.4 (57.0) | 12.7 ±0.8 (41.4) |
| Liberal Arts | 1.2 ±0.2 (86.1) | 15.8 ±1.0 (73.0) | 2.3 ±0.1 (80.9) | 9.8 ±0.6 (73.0) |
| Math | 0.9 ±0.1 (91.9) | 8.3 ±0.8 (67.3) | 1.0 ±0.3 (80.0) | 4.1 ±0.1 (67.3) |
| Math (Gen) | 4.0 ±0.3 (72.3) | 4.6 ±0.8 (28.9) | 1.5 ±0.4 (33.4) | 1.7 ±0.6 (28.9) |
| Safety/Truth | 0.9 ±0.2 (66.8) | 8.6 ±0.5 (50.0) | 3.3 ±0.2 (64.5) | 8.7 ±0.6 (50.0) |
| Science/Tech | 1.6 ±0.1 (85.0) | 12.2 ±0.5 (65.2) | 2.1 ±0.2 (76.5) | 6.4 ±0.3 (65.2) |
| **Total** | 1.3 ±0.1 (79.0) | 12.6 ±1.5 (62.0) | 2.9 ±0.2 (74.3) | 8.4 ±0.5 (62.1) |

Table 17: Trained from Instruct - High Data Scenario: Forgetting (Part 2 of 2)

| Category | Open Math | Nemotron Code Reasoning | |
| | 8B | 7B | 14B |
|---|---|---|---|
| Common Sense | 28.5 ±0.9 (64.5) | 22.1 ±0.1 (76.9) | 20.9 ±2.4 (79.0) |
| Culture | 49.9 ±1.9 (79.1) | 21.8 ±3.1 (45.3) | 20.9 ±0.7 (63.7) |
| Logic | 24.2 ±0.5 (42.5) | 14.3 ±0.4 (60.4) | 14.7 ±0.3 (75.6) |
| Knowledge/QA | 33.8 ±1.4 (60.8) | 11.9 ±1.6 (52.5) | 11.5 ±2.1 (57.9) |
| Language | 20.9 ±1.2 (39.9) | 21.2 ±1.4 (48.3) | 18.4 ±1.2 (62.4) |
| Liberal Arts | 32.1 ±2.3 (64.6) | 19.4 ±0.7 (73.0) | 12.8 ±1.0 (79.6) |
| Math | 17.1 ±1.4 (30.5) | 16.6 ±0.5 (67.3) | 13.3 ±0.4 (73.8) |
| Math (Gen) | 1.9 ±0.5 (3.7) | 11.4 ±1.2 (28.9) | 7.7 ±1.0 (33.3) |
| Safety/Truth | 19.5 ±0.7 (36.3) | 14.2 ±0.1 (50.0) | 12.3 ±1.2 (63.0) |
| Science/Tech | 24.8 ±1.8 (45.5) | 18.2 ±0.4 (65.2) | 13.0 ±0.2 (73.0) |
| **Total** | 28.8 ±0.8 (54.5) | 17.7 ±0.3 (62.4) | 14.9 ±0.0 (72.1) |

Table 18: Trained from Instruct - High Data Scenario: Backward Transfer (Part 1 of 2)

| Category | INTELLECT-2 | OpenThinker | | |
| | 32B | v1 (7B) | v2 (32B) | v3 (7B) |
|---|---|---|---|---|
| Common Sense | 1.7 ±0.1 (86.7) | 6.1 ±0.8 (60.9) | 4.2 ±0.5 (85.0) | 6.7 ±0.2 (73.2) |
| Culture | 1.1 ±0.3 (60.8) | 5.7 ±1.6 (33.6) | 2.9 ±0.3 (54.3) | 4.0 ±1.0 (23.3) |
| Logic | 0.7 ±0.1 (87.4) | 13.7 ±0.6 (67.4) | 6.2 ±0.4 (87.5) | 17.2 ±0.9 (76.3) |
| Knowledge/QA | 3.4 ±0.9 (67.5) | 7.0 ±0.7 (43.0) | 3.8 ±0.1 (62.3) | 20.0 ±0.3 (66.6) |
| Language | 1.7 ±0.2 (62.7) | 8.8 ±1.6 (39.1) | 6.4 ±0.7 (61.0) | 6.8 ±0.7 (33.6) |
| Liberal Arts | 1.2 ±0.1 (86.0) | 5.5 ±0.5 (59.2) | 5.6 ±0.3 (85.4) | 5.7 ±0.2 (67.5) |
| Math | 0.9 ±0.2 (92.1) | 11.1 ±1.0 (71.3) | 10.1 ±1.1 (91.7) | 14.7 ±0.3 (81.2) |
| Math (Gen) | 6.1 ±0.7 (74.3) | 18.9 ±0.5 (43.3) | 37.5 ±2.2 (69.5) | 38.7 ±1.6 (66.0) |
| Safety/Truth | 1.3 ±0.5 (67.2) | 7.2 ±1.2 (48.1) | 4.1 ±0.8 (65.5) | 7.5 ±0.9 (48.5) |
| Science/Tech | 1.5 ±0.1 (85.0) | 7.4 ±0.4 (58.7) | 7.6 ±0.3 (83.8) | 9.4 ±0.5 (69.2) |
| **Total** | 1.4 ±0.1 (79.2) | 7.7 ±0.5 (55.1) | 5.3 ±0.2 (77.3) | 9.7 ±0.2 (62.8) |

Table 19: Trained from Instruct - High Data Scenario: Backward Transfer (Part 2 of 2)

| Category | Open Math 8B | Nemotron Code Reasoning 7B | Nemotron Code Reasoning 14B |
|---|---|---|---|
| Common Sense | 5.7 ±0.4 (34.1) | 5.2 ±0.4 (54.3) | 3.6 ±1.2 (56.0) |
| Culture | 0.5 ±0.4 (6.5) | 5.1 ±2.3 (19.2) | 5.2 ±0.5 (37.8) |
| Logic | 6.0 ±0.4 (18.3) | 13.7 ±0.7 (57.4) | 8.1 ±0.3 (65.4) |
| Knowledge/QA | 2.8 ±1.2 (25.3) | 5.0 ±0.5 (42.6) | 3.2 ±0.2 (46.6) |
| Language | 7.2 ±0.8 (18.1) | 4.9 ±0.8 (23.3) | 7.6 ±0.8 (46.3) |
| Liberal Arts | 2.9 ±0.2 (25.5) | 4.3 ±0.1 (52.8) | 4.2 ±0.5 (68.3) |
| Math | 6.9 ±1.1 (15.0) | 10.4 ±0.0 (56.4) | 10.7 ±0.8 (69.4) |
| Math (Gen) | 14.6 ±1.4 (16.5) | 15.8 ±2.1 (33.4) | 22.0 ±1.6 (47.6) |
| Safety/Truth | 4.0 ±0.6 (15.9) | 8.6 ±1.2 (42.5) | 6.8 ±0.6 (55.6) |
| Science/Tech | 3.6 ±0.9 (17.0) | 5.2 ±0.0 (47.7) | 6.4 ±0.1 (64.3) |
| **Total** | 4.2 ±0.2 (21.1) | 6.7 ±0.4 (46.5) | 5.8 ±0.1 (59.0) |

## I.5 TRAINED FROM INSTRUCT - LOW DATA SCENARIO

Numerical results corresponding to Figure 20, covering light-touch reasoning post-training from instruction-tuned models: s1.1 (7B, 14B, 32B) and LIMO (v1 and v2).

Table 20: Trained from Instruct - Low Data Scenario: Forgetting (Part 1 of 2)

| Category | s1.1 7B | s1.1 14B | s1.1 32B |
|---|---|---|---|
| Common Sense | 8.1 ±0.4 (76.9) | 5.1 ±1.0 (82.1) | 4.7 ±0.5 (84.6) |
| Culture | 12.1 ±1.2 (45.3) | 10.6 ±0.4 (63.7) | 7.5 ±0.9 (62.1) |
| Logic | 7.2 ±0.1 (60.4) | 4.2 ±0.4 (75.6) | 2.8 ±0.2 (80.4) |
| Knowledge/QA | 5.7 ±1.1 (52.5) | 2.3 ±0.1 (57.9) | 2.3 ±1.2 (61.3) |
| Language | 8.6 ±0.8 (41.4) | 6.7 ±0.4 (54.0) | 4.9 ±0.1 (57.0) |
| Liberal Arts | 7.4 ±0.5 (73.0) | 5.4 ±0.7 (78.9) | 3.9 ±0.1 (80.9) |
| Math | 6.6 ±0.8 (67.3) | 5.8 ±0.9 (73.8) | 3.2 ±0.4 (80.0) |
| Math (Gen) | 5.7 ±0.4 (28.9) | 2.8 ±0.5 (33.3) | 1.4 ±0.1 (33.4) |
| Safety/Truth | 7.5 ±0.9 (50.0) | 5.3 ±0.3 (59.4) | 4.5 ±0.4 (64.5) |
| Science/Tech | 7.2 ±0.4 (65.2) | 4.8 ±0.6 (73.0) | 3.3 ±0.4 (76.5) |
| **Total** | 7.7 ±0.2 (62.1) | 5.3 ±0.2 (71.3) | 3.9 ±0.1 (74.3) |

Table 21: Trained from Instruct - Low Data Scenario: Forgetting (Part 2 of 2)

| Category | LIMO v1 (32B) | LIMO v2 (32B) |
|---|---|---|
| Common Sense | 3.5 ±0.3 (84.6) | 3.0 ±0.3 (84.6) |
| Culture | 3.7 ±1.9 (62.1) | 4.4 ±0.9 (62.1) |
| Logic | 3.0 ±0.3 (80.4) | 6.1 ±0.4 (80.4) |
| Knowledge/QA | 2.0 ±0.4 (61.3) | 2.4 ±0.3 (61.3) |
| Language | 3.8 ±0.6 (57.0) | 4.2 ±0.6 (57.0) |
| Liberal Arts | 2.6 ±0.0 (80.9) | 2.3 ±0.3 (80.9) |
| Math | 1.3 ±0.1 (80.0) | 1.7 ±0.1 (80.0) |
| Math (Gen) | 1.1 ±0.5 (32.9) | 1.2 ±0.3 (32.4) |
| Safety/Truth | 2.9 ±0.3 (64.5) | 2.8 ±0.2 (64.5) |
| Science/Tech | 2.3 ±0.1 (76.5) | 1.9 ±0.2 (76.5) |
| **Total** | 2.6 ±0.1 (74.3) | 3.0 ±0.2 (74.3) |

Table 22: Trained from Instruct - Low Data Scenario: Backward Transfer (Part 1 of 2)

| | s1.1 | | |
|---|---|---|---|
| Category | 7B | 14B | 32B |
| Common Sense | 7.3 ±0.5 (75.8) | 4.6 ±0.3 (81.5) | 4.5 ±0.2 (84.3) |
| Culture | 5.7 ±0.5 (35.0) | 4.6 ±0.4 (54.2) | 4.6 ±0.4 (56.7) |
| Logic | 13.8 ±0.6 (68.8) | 9.2 ±0.5 (81.1) | 5.7 ±0.4 (83.9) |
| Knowledge/QA | 13.1 ±0.3 (59.3) | 11.9 ±1.3 (69.0) | 11.3 ±0.5 (71.2) |
| Language | 10.2 ±0.7 (44.7) | 7.4 ±1.2 (55.7) | 6.6 ±0.5 (60.4) |
| Liberal Arts | 6.3 ±0.7 (71.5) | 5.3 ±0.6 (78.9) | 5.0 ±0.1 (82.4) |
| Math | 11.5 ±0.3 (74.2) | 12.0 ±0.7 (81.9) | 9.4 ±0.9 (87.8) |
| Math (Gen) | 14.3 ±0.7 (37.6) | 20.5 ±0.3 (51.0) | 32.2 ±4.3 (64.3) |
| Safety/Truth | 9.4 ±0.1 (52.5) | 7.4 ±0.9 (62.2) | 5.2 ±0.4 (65.4) |
| Science/Tech | 8.5 ±0.1 (66.8) | 7.9 ±0.3 (77.3) | 7.6 ±0.5 (82.2) |
| **Total** | 9.1 ±0.2 (63.5) | 7.2 ±0.3 (73.5) | 6.2 ±0.2 (76.9) |

Table 23: Trained from Instruct - Low Data Scenario: Backward Transfer (Part 2 of 2)

| | LIMO | |
|---|---|---|
| Category | v1 (32B) | v2 (32B) |
| Common Sense | 3.9 ±0.4 (85.1) | 4.2 ±0.2 (86.3) |
| Culture | 7.5 ±0.8 (66.1) | 6.7 ±0.8 (64.6) |
| Logic | 5.8 ±0.3 (84.6) | 5.3 ±0.6 (79.6) |
| Knowledge/QA | 11.1 ±1.3 (71.2) | 8.0 ±0.7 (67.7) |
| Language | 5.6 ±0.7 (60.3) | 5.3 ±0.3 (59.3) |
| Liberal Arts | 4.7 ±0.2 (83.7) | 4.9 ±0.2 (84.4) |
| Math | 9.6 ±0.7 (90.7) | 10.1 ±0.8 (90.8) |
| Math (Gen) | 32.6 ±4.9 (64.5) | 35.1 ±2.5 (66.4) |
| Safety/Truth | 5.0 ±0.4 (67.1) | 4.5 ±0.7 (66.7) |
| Science/Tech | 7.2 ±0.3 (83.2) | 7.5 ±0.2 (84.0) |
| **Total** | 6.2 ±0.2 (78.8) | 5.8 ±0.3 (77.9) |

## I.6 QWEN2.5 BASE AND CODER MERGE (RELATIVE TO QWEN2.5 BASE)

Numerical results for the high-drift coder merge (see Figures 25 and 26), measuring forgetting and backward transfer relative to the Qwen2.5 Base checkpoint. Both linear and Slerp interpolation produce substantial forgetting across all categories, reflecting the 87% weight-drift ratio that makes interpolation destructive.

Table 24: Qwen2.5 Base and Coder Merge: Forgetting

| | Linear ($\alpha$) | | Slerp ($\alpha$) | |
|---|---|---|---|---|
| Category | 0.2 | 0.8 | 0.2 | 0.8 |
| Common Sense | 8.2 ±0.6 (67.2) | 11.3 ±0.7 (67.2) | 13.9 ±2.2 (67.2) | 16.2 ±6.4 (67.2) |
| Culture | 12.0 ±2.0 (60.8) | 15.8 ±1.8 (60.8) | 21.1 ±3.4 (60.8) | 41.3 ±3.3 (60.8) |
| Logic | 11.6 ±0.4 (51.9) | 19.4 ±1.0 (51.9) | 23.4 ±1.1 (51.9) | 17.7 ±0.9 (51.9) |
| Knowledge/QA | 4.6 ±0.4 (56.4) | 8.1 ±0.3 (56.4) | 10.9 ±1.0 (56.4) | 16.2 ±1.0 (56.4) |
| Language | 7.8 ±0.4 (43.2) | 11.2 ±0.2 (43.2) | 14.0 ±0.5 (43.2) | 12.7 ±0.4 (43.2) |
| Liberal Arts | 2.5 ±0.5 (67.6) | 4.1 ±0.9 (67.6) | 5.8 ±1.6 (67.6) | 8.9 ±0.5 (67.6) |
| Math | 4.5 ±1.2 (44.9) | 8.9 ±2.1 (44.9) | 12.4 ±1.9 (44.9) | 9.4 ±0.7 (44.9) |
| Safety/Truth | 1.9 ±0.7 (49.9) | 3.7 ±0.6 (49.9) | 6.0 ±0.3 (49.9) | 4.7 ±1.0 (60.4) |
| Science/Tech | 2.6 ±0.6 (52.0) | 5.2 ±0.7 (52.0) | 6.7 ±1.3 (52.0) | 4.4 ±0.9 (52.0) |
| **Total** | 6.1 ±0.2 (58.0) | 9.3 ±0.4 (58.0) | 12.3 ±0.2 (58.0) | 13.9 ±0.4 (59.0) |

Table 25: Qwen2.5 Base and Coder Merge: Backward Transfer

| Category | Linear ($\alpha$) | | Slerp ($\alpha$) | |
| --- | --- | --- | --- | --- |
| | 0.2 | 0.8 | 0.2 | 0.8 |
| Common Sense | 8.3 ±0.4 (67.4) | 9.8 ±0.5 (65.2) | 9.2 ±0.2 (61.0) | 7.2 ±0.9 (55.3) |
| Culture | 4.1 ±0.8 (46.8) | 6.2 ±1.8 (46.1) | 8.3 ±2.0 (44.0) | 2.6 ±0.5 (15.1) |
| Logic | 5.3 ±0.6 (43.7) | 4.2 ±0.1 (31.0) | 3.7 ±0.1 (24.6) | 4.4 ±0.2 (33.3) |
| Knowledge/QA | 8.3 ±0.4 (61.6) | 9.2 ±0.4 (58.2) | 8.2 ±0.8 (53.4) | 5.1 ±0.6 (44.1) |
| Language | 4.2 ±0.8 (39.1) | 7.2 ±0.4 (37.4) | 6.3 ±0.2 (32.1) | 4.6 ±0.5 (30.7) |
| Liberal Arts | 2.1 ±0.4 (67.3) | 6.1 ±1.6 (70.2) | 5.7 ±2.0 (67.6) | 2.1 ±0.3 (58.7) |
| Math | 5.5 ±1.0 (45.0) | 6.5 ±1.6 (39.9) | 5.8 ±2.9 (34.0) | 4.6 ±1.4 (36.3) |
| Safety/Truth | 2.4 ±0.7 (51.8) | 5.0 ±1.9 (52.7) | 6.8 ±3.0 (51.5) | 3.0 ±1.0 (57.9) |
| Science/Tech | 2.7 ±0.3 (51.4) | 5.6 ±1.5 (52.3) | 5.2 ±1.6 (49.7) | 2.3 ±0.3 (48.1) |
| **Total** | 4.8 ±0.1 (56.0) | 6.5 ±0.7 (53.9) | 6.4 ±0.9 (49.9) | 4.0 ±0.1 (46.1) |

## I.7 QWEN2.5 BASE AND CODER MERGE (RELATIVE TO QWEN2.5 CODER)

Companion tables to I.6, measuring the same Qwen2.5 Base–Coder merge relative to the Qwen2.5 Coder checkpoint. The large forgetting figures confirm that the merged model fails to preserve Coder capabilities as well, underscoring the bidirectional damage caused by high weight drift.

Table 26: Qwen2.5 Base and Coder Merge: Forgetting

| Category | Linear ($\alpha$) | | Slerp ($\alpha$) | |
| --- | --- | --- | --- | --- |
| | 0.2 | 0.8 | 0.2 | 0.8 |
| Common Sense | 10.2 ±0.7 (70.5) | 12.7 ±0.8 (70.5) | 15.2 ±2.5 (70.5) | 18.2 ±7.0 (70.5) |
| Culture | 6.8 ±1.9 (37.8) | 10.6 ±1.4 (37.8) | 15.8 ±2.3 (37.8) | 28.5 ±3.2 (37.8) |
| Logic | 15.1 ±0.9 (47.2) | 20.4 ±0.7 (47.2) | 23.2 ±0.5 (47.2) | 18.8 ±0.3 (47.2) |
| Knowledge/QA | 6.5 ±0.8 (52.5) | 8.0 ±0.6 (52.5) | 9.8 ±0.3 (52.5) | 14.0 ±1.9 (52.5) |
| Language | 6.6 ±0.6 (29.4) | 8.1 ±0.5 (29.4) | 9.5 ±1.1 (29.4) | 9.2 ±0.9 (29.4) |
| Liberal Arts | 6.6 ±0.1 (63.3) | 6.1 ±1.0 (63.3) | 7.6 ±1.7 (63.3) | 12.3 ±0.3 (63.3) |
| Math | 17.1 ±0.5 (58.9) | 19.6 ±1.2 (58.9) | 22.2 ±2.1 (58.9) | 20.8 ±0.3 (58.9) |
| Safety/Truth | 10.9 ±1.7 (42.0) | 10.0 ±1.4 (42.0) | 10.6 ±1.3 (42.0) | 11.2 ±1.2 (43.4) |
| Science/Tech | 10.3 ±0.4 (54.3) | 10.3 ±0.9 (54.3) | 11.6 ±1.9 (54.3) | 11.9 ±0.3 (54.3) |
| **Total** | 9.5 ±0.3 (54.0) | 11.1 ±0.4 (54.0) | 13.4 ±0.3 (54.0) | 15.3 ±0.9 (54.1) |

Table 27: Qwen2.5 Base and Coder Merge: Backward Transfer

| Category | Linear ($\alpha$) | | Slerp ($\alpha$) | |
| --- | --- | --- | --- | --- |
| | 0.2 | 0.8 | 0.2 | 0.8 |
| Common Sense | 7.9 ±0.4 (67.4) | 8.7 ±0.5 (65.2) | 8.0 ±1.3 (61.0) | 6.8 ±1.6 (55.3) |
| Culture | 13.4 ±1.7 (46.8) | 15.8 ±3.2 (46.1) | 17.8 ±3.6 (44.0) | 6.2 ±1.9 (15.1) |
| Logic | 11.4 ±0.2 (43.7) | 7.5 ±0.3 (31.0) | 6.0 ±0.2 (24.6) | 7.8 ±0.0 (33.3) |
| Knowledge/QA | 15.6 ±1.0 (61.6) | 14.3 ±0.2 (58.2) | 12.2 ±1.2 (53.4) | 8.7 ±1.5 (44.1) |
| Language | 11.0 ±0.5 (39.1) | 12.2 ±0.3 (37.4) | 9.9 ±0.6 (32.1) | 9.1 ±0.2 (30.7) |
| Liberal Arts | 9.6 ±0.4 (67.3) | 11.2 ±0.9 (70.2) | 10.8 ±1.3 (67.6) | 8.5 ±0.2 (58.7) |
| Math | 7.7 ±1.0 (45.0) | 7.0 ±0.9 (39.9) | 5.1 ±1.5 (34.0) | 5.2 ±0.8 (36.3) |
| Safety/Truth | 8.7 ±1.7 (39.1) | 10.3 ±1.3 (42.4) | 9.1 ±1.7 (39.9) | 9.2 ±0.4 (41.0) |
| Science/Tech | 8.1 ±0.2 (51.4) | 8.8 ±0.6 (52.3) | 8.1 ±0.6 (49.7) | 7.2 ±0.2 (48.1) |
| **Total** | 10.0 ±0.3 (54.8) | 10.2 ±0.6 (52.9) | 9.3 ±0.7 (48.8) | 7.5 ±0.1 (44.4) |

## I.8 QWEN2.5 INSTRUCT AND OPENTHINKER 7B MERGE (RELATIVE TO QWEN2.5 INSTRUCT)

Numerical results for the moderate-drift OpenThinker merge (see Figures 27 and 28), measuring forgetting and backward transfer relative to Qwen2.5 Instruct. The low weight-drift ratio (1.8%)

translates into comparatively modest forgetting, and the Linear (0.8) interpolation yields near-zero forgetting across most categories.

Table 28: Qwen2.5 Instruct and OpenThinker 7B Merge: Forgetting

| | Linear ($\alpha$) | |
|---|---|---|
| Category | 0.2 | 0.8 |
| Logic | 8.7 ±1.5 (75.2) | 3.6 ±0.4 (75.2) |
| Knowledge/QA | 6.5 ±0.4 (60.3) | 4.2 ±2.8 (60.3) |
| Liberal Arts | 9.4 ±0.8 (73.0) | 3.7 ±0.1 (73.0) |
| Math | 10.5 ±0.6 (65.3) | 3.3 ±0.5 (65.3) |
| Safety/Truth | 7.7 ±1.3 (55.1) | 2.8 ±0.4 (55.1) |
| Science/Tech | 9.6 ±0.3 (67.4) | 4.2 ±0.2 (67.4) |
| **Total** | 8.7 ±0.2 (66.1) | 3.6 ±0.5 (66.1) |

Table 29: Qwen2.5 Instruct and OpenThinker 7B Merge: Backward Transfer

| | Linear ($\alpha$) | |
|---|---|---|
| Category | 0.2 | 0.8 |
| Logic | 6.0 ±0.7 (71.6) | 4.6 ±1.3 (76.6) |
| Knowledge/QA | 5.2 ±1.8 (58.6) | 3.8 ±0.9 (59.8) |
| Liberal Arts | 4.6 ±0.5 (66.6) | 3.9 ±0.6 (73.3) |
| Math | 8.8 ±0.3 (63.1) | 6.8 ±0.5 (70.1) |
| Safety/Truth | 5.8 ±0.3 (52.7) | 3.4 ±0.6 (56.1) |
| Science/Tech | 6.3 ±0.3 (62.9) | 4.9 ±0.3 (68.1) |
| **Total** | 6.1 ±0.1 (62.6) | 4.6 ±0.2 (67.3) |

## I.9 QWEN2.5 INSTRUCT AND OPENTHINKER 7B MERGE (RELATIVE TO OPENTHINKER)

Companion tables to I.8, measuring the same Qwen2.5 Instruct–OpenThinker merge relative to the OpenThinker checkpoint. The backward transfer values here reflect the degree to which general instruction-following capabilities are transferred into the merged model as seen from the reasoning model's perspective.

Table 30: Qwen2.5 Instruct and OpenThinker 7B Merge: Forgetting

| | Linear ($\alpha$) | |
|---|---|---|
| Category | 0.2 | 0.8 |
| Logic | 7.1 ±0.8 (64.0) | 3.7 ±1.7 (64.0) |
| Knowledge/QA | 7.3 ±1.6 (52.2) | 7.0 ±1.2 (52.2) |
| Liberal Arts | 8.5 ±0.2 (59.2) | 5.5 ±0.4 (59.2) |
| Math | 14.3 ±1.6 (72.1) | 9.4 ±1.5 (72.1) |
| Safety/Truth | 10.3 ±2.3 (52.2) | 7.2 ±1.3 (52.2) |
| Science/Tech | 9.2 ±0.2 (60.2) | 6.8 ±0.6 (60.2) |
| **Total** | 9.4 ±0.6 (60.0) | 6.6 ±0.6 (60.0) |

Table 31: Qwen2.5 Instruct and OpenThinker 7B Merge: Backward Transfer

| | Linear ($\alpha$) | |
| Category | 0.2 | 0.8 |
| --- | --- | --- |
| Logic | 12.8 ±1.6 (71.6) | 13.1 ±3.7 (76.6) |
| Knowledge/QA | 12.1 ±2.5 (58.6) | 12.6 ±2.0 (59.8) |
| Liberal Arts | 14.1 ±0.2 (66.6) | 16.1 ±0.6 (73.3) |
| Math | 7.5 ±1.5 (63.1) | 7.9 ±1.1 (70.1) |
| Safety/Truth | 10.7 ±1.0 (52.7) | 10.1 ±1.1 (56.1) |
| Science/Tech | 11.2 ±0.8 (62.9) | 12.7 ±0.4 (68.1) |
| **Total** | 11.4 ±0.7 (62.6) | 12.1 ±1.0 (67.3) |

## I.10 QWEN2.5 INSTRUCT AND OPENTHINKER3 7B MERGE (RELATIVE TO QWEN2.5 INSTRUCT)

Numerical results for the high-drift OpenThinker3 merge (see Figures 23 and 24), measuring forgetting and backward transfer relative to Qwen2.5 Instruct. The 20% weight-drift ratio drives catastrophic forgetting: values in the 30–40% range across categories reflect the output collapse described in the qualitative analysis in §F.1.

Table 32: Qwen2.5 Instruct and OpenThinker3 7B Merge: Forgetting

| | Linear ($\alpha$) | |
| Category | 0.2 | 0.8 |
| --- | --- | --- |
| Logic | 33.6 ±2.7 (75.2) | 28.1 ±1.8 (74.2) |
| Knowledge/QA | 25.7 ±8.0 (60.3) | 18.0 ±5.2 (45.0) |
| Liberal Arts | 35.0 ±2.9 (74.4) | 28.8 ±1.7 (73.8) |
| Math | 40.1 ±6.1 (65.3) | 30.0 ±0.7 (65.3) |
| Safety/Truth | 36.2 ±2.8 (64.1) | 22.8 ±3.7 (61.0) |
| Science/Tech | 30.1 ±3.2 (67.4) | 29.6 ±0.9 (67.4) |
| **Total** | 33.5 ±3.7 (67.8) | 26.0 ±0.8 (63.8) |

Table 33: Qwen2.5 Instruct and OpenThinker3 7B Merge: Backward Transfer

| | Linear ($\alpha$) | |
| Category | 0.2 | 0.8 |
| --- | --- | --- |
| Logic | 1.3 ±2.0 (31.3) | 2.3 ±1.0 (39.9) |
| Knowledge/QA | 1.9 ±1.0 (28.8) | 1.2 ±1.4 (21.7) |
| Liberal Arts | 1.9 ±0.8 (30.2) | 2.9 ±0.1 (39.2) |
| Math | 0.7 ±0.8 (12.6) | 1.6 ±0.8 (27.0) |
| Safety/Truth | 0.9 ±1.1 (15.8) | 2.9 ±0.6 (34.6) |
| Science/Tech | 2.5 ±1.0 (30.4) | 2.5 ±0.2 (30.9) |
| **Total** | 1.5 ±1.1 (24.9) | 2.2 ±0.3 (31.7) |

## I.11 QWEN2.5 INSTRUCT AND OPENTHINKER3 7B MERGE (RELATIVE TO OPENTHINKER3)

Companion tables to I.10, measuring the same Qwen2.5 Instruct–OpenThinker3 merge relative to the OpenThinker3 checkpoint. The extreme forgetting values (including Math values above 50%) confirm that the merged model fails to retain the reasoning capabilities of OpenThinker3 regardless of the interpolation coefficient.

Table 34: Qwen2.5 Instruct and OpenThinker3 7B Merge: Forgetting

| | Linear ($\alpha$) | |
| Category | 0.2 | 0.8 |
|---|---|---|
| Logic | 28.1 $\pm$3.2 (65.4) | 25.1 $\pm$2.6 (66.9) |
| Knowledge/QA | 21.2 $\pm$6.4 (54.3) | 17.2 $\pm$7.3 (38.6) |
| Liberal Arts | 31.3 $\pm$2.9 (68.9) | 27.2 $\pm$1.7 (68.3) |
| Math | 52.3 $\pm$6.9 (81.1) | 41.8 $\pm$1.8 (81.1) |
| Safety/Truth | 33.5 $\pm$2.1 (59.8) | 23.5 $\pm$1.7 (58.6) |
| Science/Tech | 32.0 $\pm$3.5 (70.7) | 32.2 $\pm$1.4 (70.5) |
| **Total** | 33.1 $\pm$3.8 (66.7) | 28.0 $\pm$1.2 (63.8) |

Table 35: Qwen2.5 Instruct and OpenThinker3 7B Merge: Backward Transfer

| | Linear ($\alpha$) | |
| Category | 0.2 | 0.8 |
|---|---|---|
| Logic | 3.0 $\pm$5.0 (31.3) | 4.9 $\pm$0.9 (39.9) |
| Knowledge/QA | 2.2 $\pm$0.7 (28.8) | 4.5 $\pm$2.3 (21.7) |
| Liberal Arts | 2.3 $\pm$0.8 (30.2) | 5.4 $\pm$0.7 (39.2) |
| Math | 0.5 $\pm$0.6 (12.6) | 1.5 $\pm$0.2 (27.0) |
| Safety/Truth | 1.0 $\pm$0.8 (15.8) | 5.5 $\pm$0.4 (34.6) |
| Science/Tech | 2.0 $\pm$0.7 (30.4) | 2.8 $\pm$0.3 (30.9) |
| **Total** | 1.8 $\pm$1.3 (24.9) | 4.0 $\pm$0.4 (31.7) |

## DISCLAIMER FOR USE OF LLMS

We primarily used LLMs in coding co-pilot applications to facilitate experimentation and help with plotting code for result presentation. LLMs were also used as writing tools to assist in refining the paper. However, the final version was carefully reviewed and finalized by the authors. No LLMs were used in ideation and experimental design.

