# OpenReview forum: "Mapping Post-Training Forgetting in Language Models at Scale"
_ICLR.cc/2026/Conference — ICLR 2026 Poster_

### Official Review · Reviewer_e6NM · 2025-10-28

**Soundness:** 2
**Presentation:** 3
**Contribution:** 1
**Rating:** 2
**Confidence:** 4

**Summary:**

This paper introduces a sample-wise framework to measure knowledge forgetting and backward transfer in language models during post-training, using chance-adjusted metrics to account for guessing in multiple-choice evaluations. Through large-scale experiments across various post-training stages, the study finds that instruction tuning and reasoning training from base models cause minimal forgetting but significant backward transfer, while domain-continual pretraining leads to moderate forgetting with limited improvement. The authors also show that model merging does not reliably mitigate forgetting, challenging common assumptions in continual learning and providing a practical tool for tracking knowledge retention in evolving AI systems.

**Strengths:**

1. The experiment is extensive
2. The research topic is realistic

**Weaknesses:**

1. Limited Conceptual Contribution and Analytical Depth
The paper reads more as an extensive experimental report than a work offering novel theoretical insights. The findings—such as "moderate forgetting" or "lower forgetting"—are described using vague, qualitative terms and do not reveal unexpected patterns or advance conceptual understanding of forgetting mechanisms in language models.

2. Methodological Rigor in Experimental Design
The evaluation framework does not adequately control for key training variables, such as learning rates or data volume across stages. For instance, observing less forgetting in SFT and RL compared to pretraining is unsurprising given the typically smaller learning rates used in later stages. Similarly, greater backward transfer in reasoning training may simply reflect the base model's initial lack of exposure to such data, rather than meaningful generalization.

**Questions:**

See weaknesses.

---

> ### Author Response · Authors · 2025-11-21
> **Rebuttal**
>
> We thank the reviewer for their time in reviewing our work and their feedback. We are glad the reviewer states our contributions in  "challenging common assumptions in continual learning" and that our experiments are extensive and topic realistic.
>
> We address the reviewer’s concerns about why our work cannot be accepted by ICLR:
>
> ---
>
> > W1.1 The paper reads more as an extensive experimental report than a work offering novel theoretical insights
>
> ICLR **explicitly welcomes empirical** as well as theoretical contributions, and our work is clearly positioned as an empirical study. A standard that implicitly requires new theory for acceptance would set an unrealistically high bar and rule out large portions of empirically driven papers published at ICLR, including several that were **candidates for the outstanding paper award**  [1].
>
> The fact that we do not introduce new theoretical results should not be the reason to undermine the work, and we hope that the reviewer agrees with us.
>
> - [1] SAM 2: Segment Anything in Images and Videos, ICLR’25
>
> ---
>
> > W1.2 The findings do not reveal unexpected patterns or advance conceptual understanding of forgetting mechanisms in language models.
>
> Our contribution is two-fold:
>
> - 1. **Conceptual understanding of forgetting:** We provide a sample-wise paradigm to address the non-fungibility of pretrained knowledge, together with clear and practical metrics.
>
> This view seems to be shared by other reviewers. E.g. Reviewer 6fZS states that “the sample-wise paradigm directly addresses the non-fungibility of pretrained knowledge (a long-overlooked limitation of task-averaged metrics), and chance-adjusted metrics rigorously correct for random guessing—an essential step for reliable evaluation of multiple-choice benchmarks (the dominant format for knowledge-intensive LM tests).”
>
> - 2. **Reveal unexpected patterns in literature:** We provide a quantitative, systematic overview of the post-training pipeline with respect to forgetting and backward transfer, uncovering surprising gaps between the continual learning literature and current practice.
>
> This interpretation, similarly, is echoed by other reviewers. Reviewer 6fZS states “the work reconciles conflicting views between continual learning literature (predicting catastrophic forgetting) and LM practice (observing minimal forgetting in many regimes), highlighting the need to adapt continual learning theories to LM-specific post-training.”
>
> We kindly request the reviewer not to simply dismiss our work as "not reveal unexpected patterns or advance conceptual understanding" with no further justification.
>
> ---
>
> > W1.3 The findings—such as "moderate forgetting" or "lower forgetting"—are described using vague, qualitative terms.
>
> In Section 3 (Experimental setup), we now clearly define the operational meanings of “low,” “moderate,” and “large” forgetting. These terms are interpretations of quantitative results, as is standard in empirical work. The main text includes plots throughout, and Appendix G reports the exact numerical values in tables. We hope this clarification addresses the concern.
>
> ---
>
> > W2.1 The evaluation framework does not adequately control for key training variables, such as learning rates or data volume across stages.
>
> Our key findings are robust across a wide range of experimental settings: different base model families (Qwen, LLaMA, Nemotron), data volumes, open-data and commercial models, training recipes, and both post-training and pretraining stages. We additionally provide, in the revised version, Appendix E which further details **how our findings vary with data diversity, objective function, and data volume**. The additional analysis shows that **data volume and domain are the most actionable** levers for understanding reasoning post-training dynamics (please refer to rebuttal to W2 to Reviewer 6fZS for details).
>
> We agree that understanding why certain effects arise is an important direction for future work, but we respectfully disagree that our work lacks “methodological rigor" for ICLR.
>
> ---
>
> > W2.2 For instance, observing less forgetting in SFT and RL compared to pretraining is unsurprising given the typically smaller learning rates used in later stages.
>
> We respectfully *disagree with the premise* that SFT and RL consistently exhibit less forgetting. In several settings **we observe substantial forgetting** in SFT/RL (see Section 3.3.3). Thus, the statement that there is “less forgetting” in these stages does not hold in many of our experiments, let alone be unsurprising. Correspondingly, smaller learning rates alone cannot explain the observed patterns.
>
> ---
>
> If the response above did not clarify your concern, could you please clarify what you meant? We are not sure if we misunderstood the concerns.
>
> We hope we have addressed the major concerns of the reviewer, and are happy to answer any further questions/concerns. We look forward to a fruitful reviewer-author discussion phase.

---

> > ### Author Response · Authors · 2025-11-28
> > **Reviewer-author Discussion**
> >
> > Dear Reviewer e6NM,
> >
> > We are still not sure if we misunderstood your concerns or whether our rebuttal addressed your major concerns. We are happy to answer further questions/concerns.
> >
> > We would be grateful if you could engage with our rebuttal, we look forward to a fruitful reviewer-author discussion.

---

### Official Review · Reviewer_wF4s · 2025-10-31

**Soundness:** 2
**Presentation:** 3
**Contribution:** 3
**Rating:** 4
**Confidence:** 3

**Summary:**

This paper addresses the core issue of knowledge change during LLM post-training, proposing an innovative and practical analytical framework. The authors first correctly identify two major shortcomings of existing average accuracy-based evaluation methods: the inability to decouple forgetting and backward transfer, and susceptibility to random guessing in multiple-choice question evaluation.

To overcome these problems, the paper proposes a sample-level metric paradigm that provides a more granular analytical perspective by tracking the transitions of individual samples from correct to incorrect and from incorrect to correct before and after training. A chance-adjusted method is introduced to model random guessing behavior, extracting the chance factor from observed transitions to estimate F_true and BT_true, which more closely approximate the actual knowledge change.

Building on this framework, the authors conducted the largest empirical study to date, systematically evaluating knowledge change across different model families and sizes in various mainstream post-training scenarios, including domain-based continuous pre-training, instruction fine-tuning, inference training, and model merging. The study yielded several important findings.

**Strengths:**

1. Shifting the evaluation perspective from the macro-level average accuracy of the task to the micro-level of the sample is a highly insightful transformation. This allows us to understand more precisely the dynamic changes in the knowledge within the model, rather than simply seeing a vague final score.

2. The authors evaluated nearly 30 model-training combinations, covering the vast majority of mainstream post-training paths in the current LLM ecosystem. This work itself is a massive undertaking, providing the community with extremely valuable data and baselines. The "forgetting map" constructed in the paper provides practitioners and researchers with a practical "benchmark" to measure the "knowledge cost" of different post-training strategies. This has direct practical value in guiding future designs for more efficient and less forgetting training processes.

**Weaknesses:**

1. The model's assumption of "uniform random guessing when unable to solve a problem, with independent guesses before and after" is somewhat idealistic. Discussions could be made regarding robustness checks using multiple sampling/deterministic decoding, or question-level chance correction based on logits, to relax the independence and uniformity assumption.

2. There are some shortcomings in the experiments. The current version reports a flipped variance/confidence interval, requiring significance and multiple comparison correction. Using random decoding with a temperature of 0.6 and a nullus value of 0.95, and running only once without multiple seed repetitions, introduces randomness leading to flipping. The variance and stability of the flipped statistics are not evaluated. The current evaluation primarily covers MCQs and has not yet been extended to tasks closer to the post-training objectives, such as open generation, code/tool ​​calls, and procedural derivation.

**Questions:**

see weakness

---

> ### Author Response · Authors · 2025-11-21
> **Rebuttal**
>
> We thank the reviewer for their time in reviewing our work and their encouraging feedback about what we feel are the core to our effort, including that the work "*allows us to understand more precisely the dynamic changes in the knowledge within the model*”, "*work itself is a massive undertaking, providing the community with extremely valuable data and baselines*” and "*has direct practical value in guiding future designs*".
>
> Below, we address the reviewer’s concerns:
>
> > W1. uniform random guessing assumption is somewhat idealistic. Discussions could be made relaxing the independence and uniformity assumption.
>
> Thank for raising this! We agree and test the validity of our chance-adjusted forgetting metric by replacing the uniform random-guessing correction with an **empirical correction based on two-seed agreement**.
>
> *Method.* Specifically, we consider only samples for which at least two seeds record the same transition in each of the four categories (0→0, 0→1, 1→0, 1→1). We then compute a samplewise forgetting metric using these two-seed-agreed transitions. This two-seed agreement acts as a filter for noisy transitions -- an empirical approximation to our uniform random-guessing correction. We then recompute forgetting using this metric and compare to chance-adjusted forgetting.
>
> *Results.* Figure 10 (https://imgur.com/a/U7DklGz) shows **the strong correlation between our original chance-adjusted forgetting metric and the two-seed agreement metric** across all models and post-training stages. Additionally, we see category-level consistency,  shown in Figure 11 (https://imgur.com/NPnq3Gt which shows the 2 seed forgetting (left) and the chance adjusted forgetting (right)), where **it displays consistent patterns across knowledge categories and along the post-training pipeline**.
>
> *Conclusion.* These additional experiments show that the **chance-adjusted metric is a reliable indicator of forgetting and backward transfer**. We hope this comparison, together with the analysis in Section C.3, clarifies the robustness of our metric and addresses major concerns  about the random-guessing assumption.
>
> ---
>
> > W2.1 Running only once without multiple seed repetitions, introduces randomness leading to flipping. The variance and stability of the flipped statistics are not evaluated.
>
> Thanks for raising this concern. We already run all experiments with three random seeds and report the standard deviation for both chance-adjusted forgetting and backward transfer in Appendix G. The standard deviations are small (typically within $\pm 0.5$–$1%$), indicating that **our forgetting and backward-transfer results are robust to randomness from seeds, temperature, and decoding**. We have updated Section 2 and Appendix B.1 to clarify this aspect.
>
> ---
>
> > W2.2 The current evaluation primarily covers MCQs and has not yet been extended to tasks closer to the post-training objectives, such as open generation ..
>
> Thanks for raising this! We perform additional experiments to convincingly demonstrate transfer to open-ended evaluation:
>
> - We expand our evaluation to six open-ended math generation benchmarks across all models: AIME24, AIME25, MATH500, AMC23, OlympiadBench, and Minerva -- a domain with clear, verifiable ground truth.
> - We compare these results to the math MCQA benchmarks to check for consistency.
>
> Concretely, all plots in Section 3 now include an additional category, “Math (Gen),” corresponding to these datasets. **We demonstrate our overall findings remain consistent**:
>
> - In Sections 3.1 (Domain-Continual Pretraining) and 3.2 (Instruction-Tuning), the dynamics for open-ended math closely mirror those of the MCQA math category, as these stages do not explicitly optimize for Math (Gen) benchmarks.
> - In Section 3.3.1, we observe the same qualitative trends as in MCQA, but with larger gains, because the models are explicitly optimized for Math (Gen) benchmarks in post-training.
> - In Section 3.3.2, we observe especially large gains for s1.1 and LIMO v2 on Math (Gen), compared to relatively small gains on MCQA. This is expected, as these models are explicitly optimized for improvements on Math (Gen) benchmarks in post-training [1].
>
> *Note on Metric.* This analysis additionally shows that our samplewise framework is directly applicable beyond MCQA. For open-ended generation tasks, chance accuracy is effectively 0, so the chance-adjustment term vanishes but **the samplewise metrics can be applied as intended to measure forgetting and backward transfer in open-ended generation** scenarios. We hope this analysis clarifies the robustness of our metric and addresses major concerns
>
> - [1] 2502.03387 -- LIMO: Less is More for Reasoning
> - [2] 2406.12624 -- Judging the Judges: Evaluating Alignment and Vulnerabilities in LLMs-as-Judges
>
> ---
>
> We hope we have addressed the major concerns of the reviewer, and are happy to answer any further questions/concerns. We look forward to a fruitful reviewer-author discussion phase.

---

> ### Author Response · Authors · 2025-11-28
> **Reviewer-author Discussion**
>
> Dear Reviewer wF4s,
>
> As the discussion period nears its conclusion, we would welcome feedback from you regarding our rebuttal.
>
> We have conducted **extensive additional analysis and experiments** to address your concerns and feedback. Specifically, we have:
>
> * Performed metric robustness tests
> * Clarified our multi-seed and variance results
> * Implemented open-ended and generative reasoning tasks
>
> Given these clarifications and new results, we have addressed the major concerns of the reviewer. If so, we would kindly ask you to consider raising your score.
>
> We thank you again for your feedback which made our work stronger and are happy to answer any further questions you may have.

---

### Official Review · Reviewer_6fZS · 2025-10-31

**Soundness:** 3
**Presentation:** 3
**Contribution:** 3
**Rating:** 6
**Confidence:** 3

**Summary:**

This paper focuses on the critical problem of pretrained knowledge forgetting and backward transfer in large-scale language model (LM) post-training, addressing the limitations of traditional evaluation methods. Unlike traditional task-averaged metrics that obscure individual knowledge changes, the paper defines forgetting as 1→0 transitions (correct before post-training, incorrect after) and backward transfer as 0→1 transitions (incorrect before, correct after), recognizing that knowledge samples (e.g., facts about U.S. presidents, API syntax) are non-fungible. To eliminate the interference of random guessing in multiple-choice benchmarks, the paper proposes \(\boldsymbol{F_{true}}\) (true forgetting) and \(\boldsymbol{BT_{true}}\) (true backward transfer). These metrics subtract the expected "chance transitions" (e.g., lucky guesses turning to errors) from raw transitions, with additional ceiling metrics (\(F_{max}\), \(BT_{max}\)) to contextualize the theoretical upper limit of knowledge change.

**Strengths:**

1-The sample-wise paradigm directly addresses the non-fungibility of pretrained knowledge (a long-overlooked limitation of task-averaged metrics), and chance-adjusted metrics rigorously correct for random guessing—an essential step for reliable evaluation of multiple-choice benchmarks (the dominant format for knowledge-intensive LM tests).

2-The methodology is both theoretically sound (e.g., explicit assumptions about uniform guessing and independent pre/post events) and practically feasible (no need for logits or repeated sampling, enabling large-scale deployment).

3-The findings provide clear practical guidance: e.g., "instruction tuning is low-risk for knowledge retention" and "large-scale RL/SFT on instruction-tuned models requires better controls"—directly informing how practitioners design post-training pipelines.

4-The work reconciles conflicting views between continual learning literature (predicting catastrophic forgetting) and LM practice (observing minimal forgetting in many regimes), highlighting the need to adapt continual learning theories to LM-specific post-training.

**Weaknesses:**

1-The paper identifies what post-training regimes cause forgetting/transfer but rarely explains why. For example:
It notes "culture" is the most forgettable category across regimes (e.g., Llama-3.1-8B-Instruct has 18.9% forgetting in culture), but does not investigate whether this stems from cultural knowledge being less "entrenched" in pretraining, or post-training data mismatching cultural contexts. It finds reasoning training on base models outperforms instruction tuning in transfer, but does not clarify if this is due to the "scratchpad" mechanism, data quality, or objective function—limiting the paper’s ability to guide how to design better post-training objectives.

2-The paper states that for RL/SFT on instruction-tuned models "at larger scales, effects are mixed and warrant further study with better controls" (1-5, 1-116), but provides no actionable path to resolve this ambiguity. It does not test variables like data diversity (e.g., mixed vs. narrow domains), training duration, or objective function (RL vs. SFT)—leaving a critical gap in understanding a common post-training scenario.

3-The paper concludes "model merging does not reliably mitigate forgetting" but only tests two-checkpoint merging (pre vs. post-training). This contrasts with prior work (e.g., Yadav et al., 2023) that uses 8+ checkpoints to mitigate weight drift. The failure to explore multi-checkpoint merging or alternative methods (e.g., TIES-merging) weakens the conclusion, as it does not rule out merging as a viable strategy—only the specific implementation tested.

4-All evaluations use MCQ formats, but real-world LM use cases often involve generative tasks (e.g., open-ended QA, text synthesis). The paper does not validate whether its sample-wise framework extends to generative settings, limiting its generalizability to practical applications where "forgetting" might manifest as ungrammaticality or factual errors in free text.

**Questions:**

Please see Weaknesses.

---

> ### Author Response · Authors · 2025-11-21
> **Rebuttal (part 1)**
>
> We thank the reviewer for their time in reviewing our work and their encouraging remarks about what we feel are the core contributions of our work, including “*directly addresses the non-fungibility of pretrained knowledge (a long-overlooked limitation of task-averaged metrics)*”, “*methodology is both theoretically sound and practically feasible*”, and “*the work reconciles conflicting views between continual learning literature…and LM practice*”.
>
> ---
>
> Below, we address the reviewer’s concerns:
>
> > W1. The paper identifies what post-training regimes cause forgetting/transfer but rarely explains why -- whether this stems from cultural knowledge being less "entrenched" in pretraining, or post-training data mismatching cultural contexts.
>
> Thank you for this feedback! For our work, the key question is whether post-training gains are primarily due to (i) improved output formatting [3,4] and (ii) better prompt-following [3,4,5], as highlighted in prior work, or instead reflect genuine changes in knowledge, such as forgetting and backward transfer. We perform experiments to isolate this below. However, addressing the full breadth of why a specific stage induces or does not induce forgetting will necessitate significant work including smart techniques to isolate the core factors, which we believe is an important direction for future work.
>
> *Method.* To better isolate knowledge forgetting from style/formatting effects known to confound results in literature, we add new experiments in Appendix C.4 using an LLM-based extraction method. The LLM answer-matcher is given the query, the model’s answer, and the ground truth, and is asked to extract the final answer. Parallel work [6] has shown this to be an effective and flexible extraction strategy. This setup reduces confounds from extraction errors or task misalignment and lets us focus more directly on forgetting and backward-transfer dynamics.
>
> *Results.* In Appendix C.4 (LLM extraction figure: https://imgur.com/a/JZCIqu2 and regex extraction figure: https://imgur.com/a/uszWy0o), **we show that our core findings remain unchanged under this improved extraction procedure**. The main difference is that some outliers shrink—for example, Qwen 2.5 Math Instruct in Figure 12. As discussed qualitatively in Section 3.1, these outliers were driven by task misalignment (e.g., coding models returning answers via print statements or variable names), where regex-based extractors often failed to recover correct answers despite the underlying output being correct.
>
> - [3] Incorrect Baseline Evaluations Call into Question Recent LLM-RL Claims, Notion Blog
> - [4] 2504.07086 -- A Sober Look at Progress in Language Model Reasoning: Pitfalls and Paths to Reproducibility
> - [5] 2506.10947 -- Spurious Rewards: Rethinking Training Signals in RLVR (Spurious prompting)
> - [6] 2507.02856 -- Answer Matching Outperforms Multiple Choice for Language Model Evaluation
>
> ---
>
> > W2. It does not test variables like data diversity (e.g., mixed vs. narrow domains), training duration, or objective function (RL vs. SFT)—leaving a critical gap in understanding a common post-training scenario.
>
> We analyze three factors: data diversity, data volume, and the training objective (SFT vs. RL). We now report all corresponding experiments explicitly in Appendix E, with links before for ease of reference.
>
> Plots:
>
> - Low data volume: https://imgur.com/a/AOLMtx4 w.r.t High data volume: https://imgur.com/a/9XprOaz
> - Narrow domains: https://imgur.com/a/0F4bqsh w.r.t Mixed domains: https://imgur.com/a/UXZvMZK
> - SFT: https://imgur.com/a/PGa8fYx w.r.t RL: https://imgur.com/a/76yEN8O
>
> Results:
>
> - **Data diversity shows a clear pattern**: Training on mixed domains consistently improves both forgetting (reduces it) and backward transfer. We now highlight this trend in the takeaway of Section 3.3.3, more explicitly in Appendix E.1.
> - **Objective does not plays a major role**: We compare SFT and RL-trained reasoning models to study the effect of the training objective. We do not find evidence that the objective plays a major role in forgetting or backward transfer (Appendix E.2).
> - **Data volumes**
>   - In the low-data regime, data volume is a good predictor of behavior: both forgetting and backward transfer are generally low, as shown in the radar plots in Section 3.3.2 and Appendix E.3.
>   - In the high-data regime, however, the results are mixed (Section 3.3.3), suggesting that while volume has a clear effect on transfer dynamics, it is not the only factor.
>
> Overall, the evidence points to **data volume and domain as the most actionable levers** for understanding reasoning post-training dynamics. We hope this provides an actionable path forward for future analysis.

---

> ### Author Response · Authors · 2025-11-21
> **Rebuttal (part 2)**
>
> > W3. The paper concludes "model merging does not reliably mitigate forgetting" but only tests two-checkpoint merging (pre vs. post-training). This contrasts with prior work (e.g., Yadav et al., 2023) that uses 8+ checkpoints to mitigate weight drift .. does not rule out merging as a viable strategy—only the specific implementation tested.
>
> Great point! We provide additional analysis to highlight **magnitude of weight drift** is a core reason for why we believe model merging is not directly viable as a strategy, explained below:
>
> *Weight drift*. We find that models whose task-vector L2 norm is large relative to the base model’s L2 norm tend to merge poorly. When this ratio exceeds roughly 10%, merge performance degrades; for example, it is 20% for OpenThinker3 but only 1.8% for OpenThinker1. This pattern is consistent with prior work: Decouple and Orthogonalize [4] links large magnitude differences between task vectors, empirically and theoretically, degrade merging performance. TIES-Merging [3] touches upon a similar discussion. We include a discussion about weight drift in Appendix D.1 as the primary cause for why merging is far harder.
>
> - [3] 2306.01708 -- TIES-Merging: Resolving Interference When Merging Models
> - [4] 2505.15875 -- Decouple and Orthogonalize: A Data-Free Framework for LoRA Merging
>
> We are in the process of running experiments using TIES and Linear+DARE -- intial results confirm that techniques for reducing interference does not help with the core issue. We will update the draft with full results soon!
>
> *Note regarding 2 vs 8 models*. Past work including TIES-Merging and Yadav et al., 2023 clearly state that merging more models is harder due to more interference, merging two models (our case) is the easiest. In the case of only two available models along a training pipeline (e.g. base to coder model), to our best knowledge, linear interpolation remains the standard technique.
>
> ---
>
> > W4. The paper does not validate whether its sample-wise framework extends to generative settings
>
> Thanks for raising this! We clarify two aspects:
>
> 1. Our metric is not restricted to MCQA benchmarks. For open-ended generation tasks, chance accuracy is effectively 0, so the chance-adjustment term vanishes but the samplewise metrics can be applied as intended beyond MCQA to open-ended generation benchmarks.
>
> 2. We perform additional experiments to convincingly demonstrate transfer to open-ended evaluation:
>
> - We expand our evaluation to six open-ended math generation benchmarks across all models: AIME24, AIME25, MATH500, AMC23, OlympiadBench, and Minerva -- a domain with clear, verifiable ground truth.
> - We compare these results to the math MCQA benchmarks to check for consistency.
>
> Concretely, all plots in Section 3 now include an additional category, “Math (Gen),” corresponding to these datasets. **We demonstrate our overall findings remain consistent**:
>
> - In Sections 3.1 (Domain-Continual Pretraining) and 3.2 (Instruction-Tuning), the dynamics for open-ended math closely mirror those of the MCQA math category, as these stages do not explicitly optimize for Math (Gen) benchmarks.
> - In Section 3.3.1, we observe the same qualitative trends as in MCQA, but with larger gains, because the models are explicitly optimized for Math (Gen) benchmarks in post-training.
> - In Section 3.3.2, we observe especially large gains for s1.1 and LIMO v2 on Math (Gen), compared to relatively small gains on MCQA. This is expected, as these models are explicitly optimized for improvements on Math (Gen) benchmarks in post-training [1].
>
> Note on MCQA benchmarks. We focus on MCQA benchmarks as **this format dominates knowledge-based benchmarks**, provides unambiguous ground truth, and avoids many reliability issues associated with open-ended evaluation (e.g., instability in LLM-as-a-judge settings [2]). This lets us more cleanly identify the sources of forgetting and backward transfer, while the new Math (Gen) results show that our conclusions extend beyond MCQA.
>
> - [1] 2502.03387 -- LIMO: Less is More for Reasoning
> - [2] 2406.12624 -- Judging the Judges: Evaluating Alignment and Vulnerabilities in LLMs-as-Judges
>
> ---
>
> We hope we have addressed the major concerns of the reviewer, and are happy to answer any further questions/concerns. We look forward to a fruitful reviewer-author discussion phase.

---

> ### Author Response · Authors · 2025-11-28
> **Reviewer-author Discussion**
>
> Dear Reviewer 6fZS,
>
> As the discussion period nears its conclusion, we would welcome feedback from you regarding our rebuttal.
>
> We have conducted **extensive additional analysis and experiments** to address your concerns and feedback. Specifically, we have:
> - Isolated data diversity, data volume, and the training objective to find the most actionable levers
> - Improved extraction procedure with LLM answer matching
> - Investigated why merging might fail -- i.e. large magnitude of weight drift
> - Implemented open-ended and generative reasoning tasks.
>
> Given these clarifications and new results, we have addressed the major concerns of the reviewer. If so, we would kindly ask you to consider raising your score.
>
> We thank you again for your feedback which made our work stronger and are happy to answer any further questions you may have.

---

### Official Review · Reviewer_f1yS · 2025-11-02

**Soundness:** 3
**Presentation:** 3
**Contribution:** 3
**Rating:** 8
**Confidence:** 3

**Summary:**

This study introduces a sample-wise framework to quantify how LLMs forget or retain knowledge during post-training. By measuring 1→0 (forgetting) and 0→1 (backward transfer) transitions, it finds that instruction tuning and reasoning post-training improve capabilities with minimal forgetting, while domain-continual pretraining causes moderate forgetting.

**Strengths:**

- This paper introduces a sample-wise, chance-adjusted metric to precisely quantify forgetting and backward transfer, overcoming the limitations of aggregate accuracy measures.
- This paper conducts a broad empirical studies across multiple model sizes, training regimes (SFT, RLHF, instruction tuning, continual pretraining), and datasets, providing a systematic view of post-training dynamics.
- This paper offers interest findings: instruction tuning and reasoning post-training yield strong backward transfer with little forgetting, while domain-continual pretraining induces moderate loss.
- This work challenges the assumption of “catastrophic forgetting” in post-trained LLMs, showing that most modern post-training does not erode pretrained knowledge as severely as previously thought.

**Weaknesses:**

- This study focuses mainly on MCQ datasets, which may not generalize to open-ended tasks or generative reasoning.
- The uniform random-guessing correction assumes independence and equal likelihood of options, which oversimplifies model behavior and may distort real forgetting dynamics.
- While correlations between data scale, post-training type, and forgetting are documented, the paper does not deeply analyze why certain stages (e.g., reasoning training) cause particular effects.

**Questions:**

N/A

---

> ### Author Response · Authors · 2025-11-21
> **Rebuttal (part 1)**
>
> We thank the reviewer for their time in reviewing our work and their encouraging remarks about what we feel are the core contributions of our work, including our study “*overcoming the limitations of aggregate accuracy measures*” and providing “*broad empirical studies*” that challenge the assumption of “*post-training not erode pretrained knowledge as severely as previously thought*”.
>
> ---
>
> Below, we address the reviewer’s concerns:
>
> > *W1. Study on MCQ datasets might not generalize to open-ended tasks or generative reasoning.*
>
> Thanks for raising this! We clarify two aspects:
>
> 1. Our metric is not restricted to MCQA benchmarks. For open-ended generation tasks, chance accuracy is effectively 0, so the chance-adjustment term vanishes but the samplewise metrics can be applied as intended beyond MCQA to open-ended generation benchmarks.
>
> 2. We perform additional experiments to convincingly demonstrate transfer to open-ended evaluation:
>
> - We expand our evaluation to six open-ended math generation benchmarks across all models: AIME24, AIME25, MATH500, AMC23, OlympiadBench, and Minerva -- a domain with clear, verifiable ground truth.
> - We compare these results to the math MCQA benchmarks to check for consistency.
>
> Concretely, all plots in Section 3 now include an additional category, “Math (Gen),” corresponding to these datasets. **We demonstrate our overall findings remain consistent**:
>
> - In Sections 3.1 (Domain-Continual Pretraining) and 3.2 (Instruction-Tuning), the dynamics for open-ended math closely mirror those of the MCQA math category, as these stages do not explicitly optimize for Math (Gen) benchmarks.
> - In Section 3.3.1, we observe the same qualitative trends as in MCQA, but with larger gains, because the models are explicitly optimized for Math (Gen) benchmarks in post-training.
> - In Section 3.3.2, we observe especially large gains for s1.1 and LIMO v2 on Math (Gen), compared to relatively small gains on MCQA. This is expected, as these models are explicitly optimized for improvements on Math (Gen) benchmarks in post-training [1].
>
> Note on MCQA benchmarks. We focus on MCQA benchmarks as **this format dominates knowledge-based benchmarks**, provides unambiguous ground truth, and avoids many reliability issues associated with open-ended evaluation (e.g., instability in LLM-as-a-judge settings [2]). This lets us more cleanly identify the sources of forgetting and backward transfer, while the new Math (Gen) results show that our conclusions extend beyond MCQA.
>
> - [1] 2502.03387 -- LIMO: Less is More for Reasoning
> - [2] 2406.12624 -- Judging the Judges: Evaluating Alignment and Vulnerabilities in LLMs-as-Judges
>
> ---
>
> > *W2. The uniform random-guessing correction might oversimplify model behavior and may distort real forgetting dynamics.*
>
> Thank you for raising this! We have added a new analysis in Section C.3 (“Metric Robustness under MCQA”).
>
> We test the validity of our chance-adjusted forgetting metric by replacing the uniform random-guessing correction with an **empirical correction based on two-seed agreement**. Specifically, we consider only samples for which at least two seeds record the same transition in each of the four categories (0→0, 0→1, 1→0, 1→1). We then compute a samplewise forgetting metric using these two-seed-agreed transitions. This two-seed agreement acts as a filter for noisy transitions -- an empirical approximation to our uniform random-guessing correction. We then recompute forgetting using this metric and compare to chance-adjusted forgetting.
>
> Figure 10 (https://imgur.com/a/U7DklGz) shows **the strong correlation between our original chance-adjusted forgetting metric and the two-seed agreement metric** across all models and post-training stages. Additionally, we see category-level consistency,  shown in Figure 11 (https://imgur.com/NPnq3Gt which shows the 2 seed forgetting (left) and the chance adjusted forgetting (right)), where **it displays consistent patterns across knowledge categories and along the post-training pipeline as chance-adjusted metrics**.
>
> These additional experiments show that the **chance-adjusted metric is a reliable indicator of forgetting and backward transfer**. We hope this comparison, together with the analysis in Section C.3, clarifies the robustness of our metric and addresses your concern about the random-guessing assumption.

---

> ### Author Response · Authors · 2025-11-21
> **Rebuttal (part 2)**
>
> > W3. While correlations between data scale, post-training type, and forgetting are documented, the paper could more deeply analyze why certain stages (e.g., reasoning training) cause particular effects.
>
> Thank you for this feedback. For our work, the key question is whether post-training gains are primarily due to (i) improved output formatting [3,4] and (ii) better prompt-following [3,4,5], as highlighted in prior work, or instead reflect genuine changes in knowledge, such as forgetting and backward transfer, which are our focus.
>
> To better isolate these effects, we add new experiments in Appendix C.4 using an LLM-based extraction method. The LLM answer-matcher is given the query, the model’s answer, and the ground truth, and is asked to extract the final answer. Parallel work [6] has shown this to be an effective and flexible extraction strategy. This setup reduces confounds from extraction errors or task misalignment and lets us focus more directly on forgetting and backward-transfer dynamics.
>
> In Appendix C.4 (LLM extraction figure: https://imgur.com/a/JZCIqu2 and  regex extraction figure: https://imgur.com/a/uszWy0o), **we show that our core findings remain unchanged under this improved extraction procedure**. The main difference is that some outliers shrink—for example, Qwen 2.5 Math Instruct in Figure 12. As discussed qualitatively in Section 3.1, these outliers were driven by task misalignment (e.g., coding models returning answers via print statements or variable names), where regex-based extractors often failed to recover correct answers despite the underlying output being correct.
>
> A detailed analysis of why a specific stage induces or does not induce forgetting will necessitate significant work including smart methods to isolate the core factors, which we believe is an important direction for future work.
>
> - [3] Incorrect Baseline Evaluations Call into Question Recent LLM-RL Claims, Notion Blog
> - [4] 2504.07086 -- A Sober Look at Progress in Language Model Reasoning: Pitfalls and Paths to Reproducibility
> - [5] 2506.10947 -- Spurious Rewards: Rethinking Training Signals in RLVR (Spurious prompting)
> - [6] 2507.02856 -- Answer Matching Outperforms Multiple Choice for Language Model Evaluation
>
> ---
>
> We hope we have addressed the major concerns of the reviewer, and are happy to answer any further questions/concerns. We look forward to a fruitful reviewer-author discussion phase.

---

> > ### Author Response · Authors · 2025-11-28
> > **Reviewer-author Discussion**
> >
> > Dear Reviewer f1yS,
> >
> > As the discussion period nears its conclusion, we would welcome feedback from you regarding our rebuttal.
> >
> > We have conducted **extensive additional analysis and experiments** to address your concerns and feedback. Specifically, we have:
> >
> > * Implementation of open-ended and generative reasoning tasks.
> > * Metric robustness tests.
> > * Improved extraction procedure with LLM answer matching
> >
> > We hope we have addressed the major concerns of the reviewer, and we thank you again for your feedback which made our work stronger.
> >
> > We are happy to answer any further questions you may have.

---

### Meta-Review · Area_Chair_NC8T · 2025-12-19

**Summary:**

1. The uniform random guessing is too simple or idealistic. (Reviewer f1yS and wF4s)
2. This paper mainly conducts evaluations on MCQ datasets, missing other generative benchmarks. (Reviewer f1yS, 6fZS, and wF4s)
3. Lacks deep analysis: While correlations between data scale, post-training type, and forgetting are documented, the paper does not deeply analyze why certain stages (e.g., reasoning training) cause particular effects. (Reviewer f1yS and 6fZS)
4. It does not test variables like data diversity (e.g., mixed vs. narrow domains), training duration, or objective function (RL vs. SFT)—leaving a critical gap in understanding a common post-training scenario. (Reviewer 6fZS)

**Reviewer Concerns:**

The authors have addressed almost all the reviewers' questions during the rebuttal phase. It is recommended to incorporate these experimental results into the new version and further refine the paper to improve its quality.

**Reviewer Scores:**

Reviewer f1yS: retains 8

Reviewer 6fZS: 6 --> 8

Reviewer wF4s: 4 --> 6

Reviewer e6NM: 2 --> 4

---

### Decision · Program_Chairs · 2026-01-26

Accept (Poster)